# *Drosophila* mechanical nociceptors preferentially sense localized poking

**Zhen Liu[1†], Meng-Hua Wu[1†], Qi-Xuan Wang[1], Shao-Zhen Lin[2], Xi-Qiao Feng[2], Bo Li[2], Xin Liang[1]\***

[1]School of Life Sciences, Tsinghua University, Beijing, China; [2]Institute of Biomechanics and Medical Engineering, Department of Engineering Mechanics, Tsinghua University, Beijing, China

**Abstract** Mechanical nociception is an evolutionarily conserved sensory process required for the survival of living organisms. Previous studies have revealed much about the neural circuits and sensory molecules in mechanical nociception, but the cellular mechanisms adopted by nociceptors in force detection remain elusive. To address this issue, we study the mechanosensation of a fly larval nociceptor (class IV da neurons, c4da) using a customized mechanical device. We find that c4da are sensitive to mN-scale forces and make uniform responses to the forces applied at different dendritic regions. Moreover, c4da showed a greater sensitivity to localized forces, consistent with them being able to detect the poking of sharp objects, such as wasp ovipositor. Further analysis reveals that high morphological complexity, mechanosensitivity to lateral tension and possibly also active signal propagation in dendrites contribute to the sensory features of c4da. In particular, we discover that Piezo and Ppk1/Ppk26, two key mechanosensory molecules, make differential but additive contributions to the mechanosensitivity of c4da. In all, our results provide updates into understanding how c4da process mechanical signals at the cellular level and reveal the contributions of key molecules.

## Editor's evaluation

Liu et al. present a fascinating study that significantly advances fundamental knowledge about the molecular and cellular pathways underlying mechanical nociception. The use of a combination of fine biophysics and neurogenetics provides unprecedented insight into mechanosensory functions in an intact tissue environment of the *Drosophila* larva. The results of this work have strong implications for our understanding of the sensation of acute pain.

**\*For correspondence:**
xinliang@tsinghua.edu.cn

[†]These authors contributed equally to this work

**Competing interest:** The authors declare that no competing interests exist.

## Introduction

Mechanosensation is a physiological process that transduces mechanical stimuli into neural signals (*Chalfie, 2009*). It underlies the perception of gentle touch, sound, acceleration and noxious force. To cope with manifold environmental forces, mechanoreceptor cells are diversified (*Lumpkin et al., 2010*; *Zimmerman et al., 2014*). Among different types of mechanoreceptor cells, those activated by noxious forces, that is mechanical nociceptors, are of particular importance because they are one of the sensory organs that are essential for the survival of living organisms (*Lumpkin et al., 2010*; *Tracey, 2017*).

Much effort has been made to understand mechanical nociception in several model organisms. In mammals, free nerve endings of nociceptive neurons penetrate the keratinocyte layer of skin and serve as the primary nociceptors. The axons of these neurons output to neural circuits in the spinal cord, which then transmit pain signals to local interneurons or up to the brain to initiate neural reflexes (*Tracey, 2017*). At the molecular level, transient receptor potential (TRP) channels, Piezo

**eLife digest** Being able to sense harm is essential for survival. Animals have to be able to tell the difference between a gentle touch and a dangerous pressure. They do this using nerve cells called mechanical nociceptors which switch on when the body feels a potentially painful pressure, such as a sharp object poking the skin. Once activated, the nerves send outputs to other parts of the central nervous system which coordinate the motions needed to escape the source of the pain.

One popular model to understand harm-sensing is the larvae of fruit flies which automatically roll back and forth when they sense the pointy sting of a wasp. This process is initiated by sensory nerve cells called class IV dendritic arborization neurons (or c4da for short) which sit under the fly's skin. However, it is still not fully understood how these mechanical nociceptors detect the poking forces of the wasp's tail.

To investigate, Liu, Wu et al. built a device that could poke sections of fly larvae under a microscope so they could see how different types of pressure affected the activity and shape of c4da cells. This revealed that c4da nerves were most sensitive to sharp objects that illicit a more localized force, which may explain why these cells are so good at responding to wasp attacks.

Further analysis showed that this sensitivity was due to the high number of branches, or dendrites, protruding from the body of c4da nerves. Liu, Wu et al. discovered that the dendrites were coated in a touch-sensitive protein that can sense and amplify both squashing and pulling, resulting in a signal that activates c4da nerves to send outputs to other parts of the central nervous system. This mechanism increases the likelihood that a c4da cell will detect a mechanical pressure even if it is far away from the body of the nerve.

These findings shed light on how sensory cells like c4da are optimized to carry out specific roles. This could be important for understanding other nerve systems which sense mechanical pressure, such as those involved in touch or auditory processes. However, further work is needed to see whether the molecules and mechanism identified by Liu, Wu et al. are also present in humans.

and other channels are found to be involved in mechanical nociception (*Kwan et al., 2006*; *Tracey, 2017*; *Murthy et al., 2018*). In invertebrates (such as worms and flies), mechanical nociception has also been extensively studied (*Chatzigeorgiou et al., 2010*; *Tracey, 2017*). A widely used model is fly larval mechanical nociception mediated by class IV dendritic arborization neurons (c4da). C4da are polymodal nociceptors that can be activated by light, thermal and mechanical stimuli (*Hwang et al., 2007*; *Xiang et al., 2010*; *Kim et al., 2012*; *Terada et al., 2016*). The axon of c4da synapses to several targets in the ventral nerve cord, which then output signals to trigger the rolling behavior, a stereotyped locomotion in escaping from nociceptive stimuli (*Grueber et al., 2007*; *Ohyama et al., 2015*; *Gerhard et al., 2017*; *Hu et al., 2017*; *Kaneko et al., 2017*; *Takagi et al., 2017*; *Yoshino et al., 2017*; *Burgos et al., 2018*). Furthermore, two parallel molecular pathways in c4da, one mediated by Ppk1/Ppk26 and the other by Piezo, are proposed to be responsible for the behavioral responses in mechanical but not thermal nociception (*Kim et al., 2012*; *Zhong et al., 2010*; *Gorczyca et al., 2014*; *Guo et al., 2014*).

An intriguing question is whether c4da neurons are optimized at the cellular level for its function as mechanical nociceptor? A recent study shows that the nociceptive response of c4da to thermal stimuli depends on the dendritic calcium influx through two TRPA channels and the L-type voltage-gated calcium channel, suggesting the presence of neuronal processing of heat-induced responses (*Terada et al., 2016*). In a more general sense, it is also intriguing how mechanoreceptor cells are optimized for their specific stimuli. An interesting study reports unique neuronal mechanisms in the mechanosensory neurons in tactile-foraging birds (*Schneider et al., 2014*), also demonstrating the presence of cellular mechanisms that facilitate the response to specific stimuli. So far, how c4da process mechanical inputs in nociception at the neuronal level remains unclear.

To address this issue, we build a 'mechanical-optical' recording system that is able to measure the in vivo sensory response of c4da to controlled forces. We find that c4da are sensitive to mN-scale forces and use the entire dendritic territory (including the non-neuronal part of the epidermis) as the force-receptive field. In particular, c4da show greater responses to smaller probes, suggesting their sensory preference to localized forces. Further analysis reveals the cellular mechanisms that facilitate

the sensory features of c4da and the contributions of key molecules. In all, these findings update the current model for the c4da-mediated mechanical nociception and provide novel insights into how mechano-nociceptor cells process force signals at the cellular level.

## Results

### Mechanical recording and the 'sphere-surface' contact model

We set out by building a mechanical device that was able to exert and measure compressive forces onto fly larval fillets (*Figure 1A*). This device contained three parts: a piezo stack actuator, a metal beam coupled with a strain gauge and a glass force probe (*Figure 1A* and *Figure 1—figure supplement 1*). The whole device was mounted on the working stage of an inverted spinning-disk confocal microscope (*Figure 1A* and *Figure 1—figure supplement 1*). To record the force-evoked response of sensory neurons, freshly prepared larval fillet was spread and mounted on a polydimethylsiloxane (PDMS) pad (thickness: 1 mm) with the exterior surface of the cuticle accessible to the force probe and the interior surface visible to the confocal microscope (*Figure 1A*). When the force probe, driven by the piezo actuator, delivered compressive forces onto the larval fillet, the strain gauge converted the deflection of the metal beam into an electrical signal. The electrical signal could be translated into a force signal using the calibration curve (*Figure 1B–C*, see **Materials** and **methods**). By making spherical probes in different diameters (*Figure 1D*), we can study how mechanoreceptor cells respond to the probes of different sizes. In the meantime, we also monitored the position of force probes, dendritic morphology (membrane marker) and neuronal response (GCaMP6s) (*Figure 1E*).

The contact mechanics between the force probe and fillet can be approximated using a classic 'sphere-surface' contact model (*Johnson, 1985*), which would allow us to separate the effects of different parameters, such as pressure and contact area. Note that to guarantee the accuracy of the contact model, the indentation depth ($d$) needs to be smaller than the radius of the spherical probe ($r$) (*Figure 1F*). This prerequisite is acceptable for our measurements because our pilot experiments showed that when the step distance ($D$) of the piezo actuator was greater than $r$, the force probe often penetrated the cuticle and caused tissue damage. The small deformation constraint would naturally minimize tissue damage (*Figure 1—figure supplement 1*) and allow us to focus on mechanical nociception (i.e. no major interference from chemo-nociception due to tissue damage). To confirm if the modeling approximation is valid, we calculated contact force ($f_{model}$) from $d$ (*Equations 1 and 2*, see Materials and methods), which could be obtained from the measured values of $D$ and beam deflection ($B$) (*Equation 3*, see Materials and methods). $f_{model}$ was then compared to the experimentally measured forces ($f_{exp}$) (*Figure 1G–I*). As shown in *Figure 1I*, the relationship between $f_{exp}$ and $d$ fell in the range of model calculations, demonstrating that the model approximation is valid. Note that different values for the elastic modulus of PDMS ($E_{PDMS}$) (*Johnston et al., 2014*; *Vlassov et al., 2018*) were used to calculate the upper and lower bounds of contact forces (*Figure 1I*).

### C4da are sensitive to mN-scale forces and more sensitive to small probes

Using the customized mechanical device, we applied poking forces onto c4da at the positions of about 100 μm away from the soma (i.e. the 'proximal' region, denoted as **p** in the plots) (*Figure 2A* and *Video 1*). By simultaneously monitoring the position of the probes and neuronal morphology, we ensured that the probe made direct compression onto the dendrites. Force titration experiments using a 60 μm probe showed a half-activation force ($f_{50}$) of ~3 mN and a full-activation force ($f_{90}$) of ~4 mN, while those using a 30 μm probe showed a $f_{50}$ of ~0.7 mN and a $f_{90}$ of ~1.5 mN (*Figure 2B–C*). This observation suggests that the mechanosensitivity of c4da is dependent on probe size. To explore the role of the entire dendritic field in force detection, we compressed the dendrites of c4da at the positions of about 200 μm away from the soma (i.e. the 'distal' region, denoted as **d** in the figures) (*Figure 2D*). When the saturation force (4 mN) of the 60 μm probe was used, the responses of c4da were not significantly different from those to the proximal stimuli (*Figure 2E*). In the case of using a 30 μm probe and applying the saturation force (1.5 mN), the responses of c4da were to some extent weaker than those to the proximal stimuli, but most of the neurons still made clear responses (*Figure 2E*). Therefore, our results showed that c4da could respond to the forces applied onto any part of their dendritic arbors.

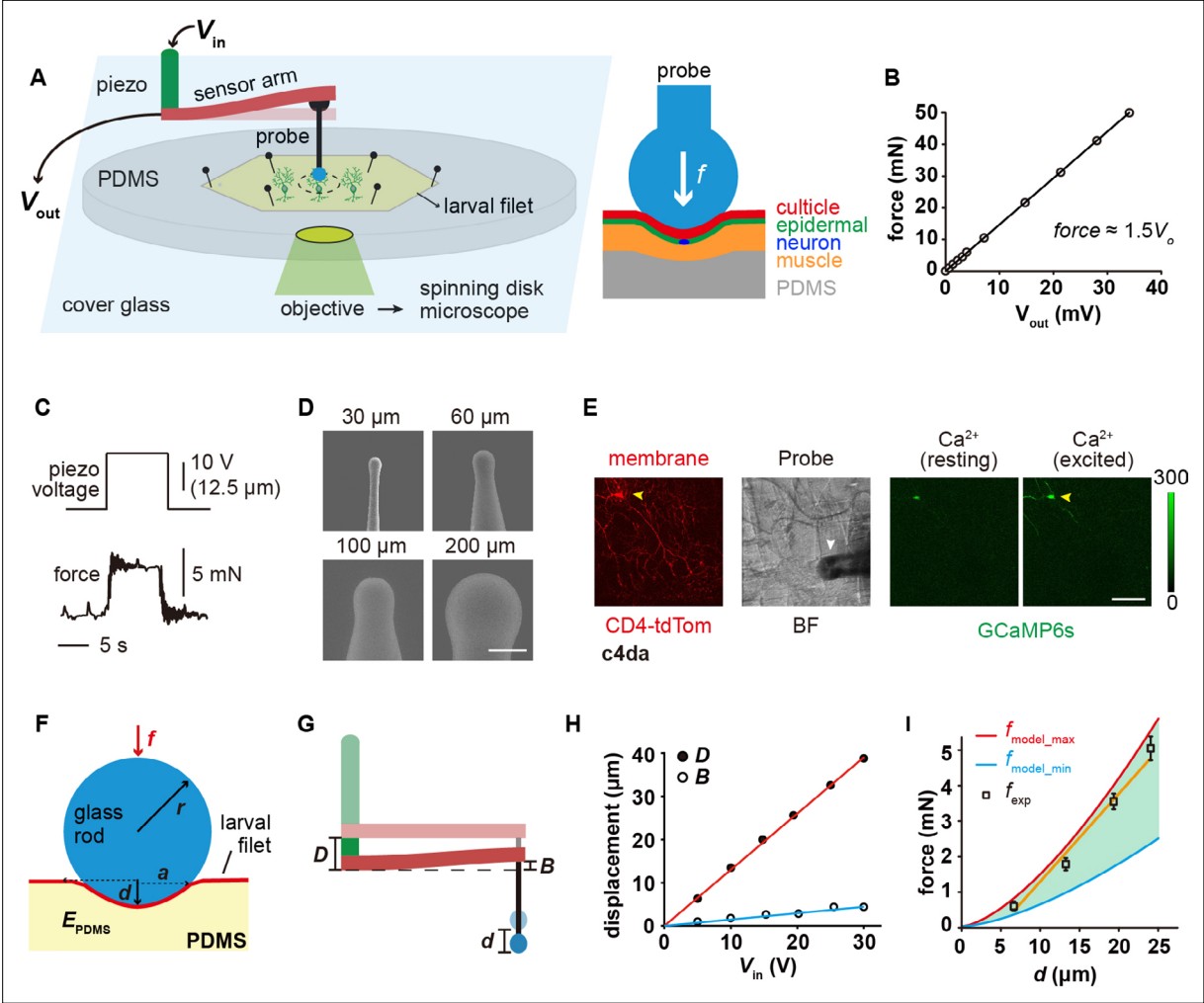

**Figure 1.** The 'mechanical-optical' recording system. (**A**) The cartoon schematics for the 'mechanical-optical' recording system (left) and the contact model between a spherical probe and the larval fillet (right). $V_{in}$ was the driving voltage of the piezo actuator. $V_{out}$ was the readout voltage of the strain gauge. (**B**) The force calibration curve of the strain gauge. The data points were mean values from three measurements. (**C**) Representative traces for the input (driving voltage of the piezo actuator, upper panel) and the mechanical output (stimulating force, lower panel) of the recording system. (**D**) The scanning electron microscopy images of glass force probes of different sizes. Scale bar, 100 μm. (**E**) Left panel: a representative image of c4da (membrane, red channel), Middle panel: a bright field image of larval fillet. Right panel: two representative images showing the GCaMP6s signals in c4da at resting (left) and exciting (right) conditions. Yellow arrowhead: soma. White arrowhead: force probe. Scale bar, 100 μm. Genotype: *uas-cd4-tdTom; ppk-gal4/20×uas-ivs-gcamp6s*. (**F**) The mechanical schematics of the 'sphere-surface' model. Note that the contact interface had a spherical crown shape. The definitions of all model parameters were described in **Materials** and **methods**. (**G**) The cartoons schematics showing the relationship among indentation depth of the force probe (**d**) deflection of the beam (**B**) and stepping distance of the piezo (**D**). (**H**) The plots of D (red) and B (blue) versus the driving voltage of the piezo ($V_{in}$). (**I**) The comparison between the calculated (red and blue) and experimentally measured (black) contact forces. In our calculations, the maximal (4 MPa, red) and minimal (1.6 MPa, blue) values of the elastic modules of PDMS were from the literatures (*Johnston et al., 2014*; *Vlassov et al., 2018*). Data were presented as mean ± std (n=9 assays).

The online version of this article includes the following source data and figure supplement(s) for figure 1:

**Source data 1.** Numerical data for *Figure 1*.

**Figure supplement 1.** The customized 'mechanical-optical' recording system.

To further test the validity of this assay, we performed the same assay on c1da (class I da neuron) and c3da (class III da neuron). C1da showed no responses to compressive forces (*Figure 2—figure supplement 1*), consistent with their function being a proprioceptor rather than a tactile receptor (*Hughes and Thomas, 2007*; *Guo et al., 2016*; *He et al., 2019*). C3da showed a much smaller threshold of activation, suggesting a higher sensitivity in detecting tactile forces (using both 30 and 60 μm probes).

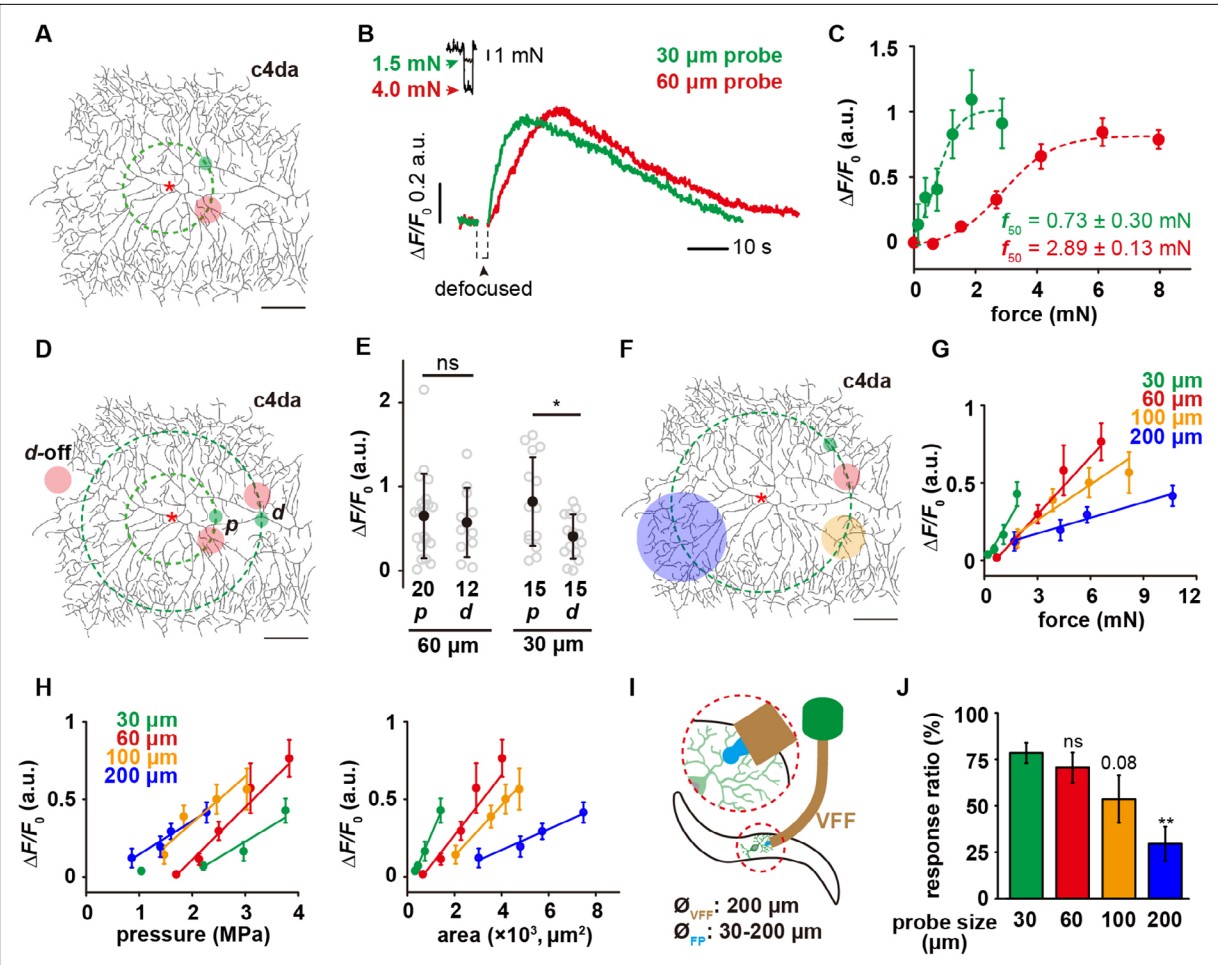

**Figure 2.** The mechanosensory features of c4da. (**A**) A representative image of c4da. The forces were applied at about 100 μm from the soma, i.e. along the green dashed circle. The representative force application points were marked using the filled circles (Green: 30 μm probe. Red: 60 μm probe). Genotype: *uas-cd4-tdtom; ppk-gal4.* (**B**) Representative responses of c4da ($\Delta F/F_0$, i.e. the change in calcium signal in the soma, unless otherwise stated hereinafter) to mN-scale forces delivered using the 30 μm (green) and 60 μm (red) probes. The black arrowhead indicated the defocused period of the soma caused by the stimulating force (2 s). Genotype: *ppk-gal4/+; ppk-cd4-tdtom/20×uas-ivs-gcamp6s.* (**C**) The force-response ($\Delta F/F_0$) plots of c4da (n=12 cells). The dashed lines were Boltzmann fitting. (**D**) The schematic showing the force application points (filled circles, green: 30 μm probe, red: 60 μm probe) of different stimuli. The dashed concentric circles were 100 and 200 μm in radius, respectively. (**p**) proximal dendrite. (**d**) distal dendrite. *d*-off: the 'dendrite-off' region. (**E**) The responses of c4da ($\Delta F/F_0$) to the proximal and distal stimuli using the 30 (1.5 mN) and 60 μm (4 mN) probes. Mann Whitney U test was used. \*: p<0.05. ns: no significance. (**F**) The schematic showing the stimuli (filled circles, green: 30 μm probe, red: 60 μm probe, orange: 100 μm probe, blue: 200 μm probe) delivered using the probes of different sizes. The dashed circle was 200 μm in radius. (**G**) The responses of c4da ($\Delta F/F_0$, n=10 cells) to the forces applied on the distal dendrites using the probes of different sizes. (**H**) The plots of the responses of c4da (the same dataset as panel (**G**)) to distal stimuli versus central pressure ($P_0$) (left panel) and contact areas ($A_c$) (right panel), respectively. Also see *Figure 2—figure supplement 2* for the plots of the responses versus the pressures at other positions ($P_{x\ μm}$). (**I**) The cartoon schematics of the modified behavior assay for mechanical nociception. VFF: Von Frey fiber, FP: force probe. (**J**) The behavioral responses of wild-type larvae to the mechanical poking stimuli using the probes of different sizes. The experiments were performed three times and the total numbers of larvae used for each type of probe were as following: 30 μm probe (n=92), 60 μm probe (n=67), 100 μm probe (n=69), 200 μm probe (n=70). One-way ANOVA test was used. \*\*: p<0.01. ns: no significance. In panels (**A**), (**D**) and (**F**), scale bar: 100 μm. Asterisk: the soma. In panels (**C**), (**G**) and (**H**), data were presented as mean ± sem. In panel (**E**) and (**J**), data were presented as mean ± std and the numbers of cells were indicated below the scattered data points.

The online version of this article includes the following source data and figure supplement(s) for figure 2:

**Source data 1.** Numerical data for *Figure 2*.

**Figure supplement 1.** Mechanosensory responses of c1da and c3da.

**Figure supplement 1—source data 1.** Numerical data for *Figure 2—figure supplement 1*.

**Figure supplement 2.** The responses of c4da-wt to distal stimuli have a linear scaling relationship with the pressure ($P_{x\ μm}$).

*Figure 2 continued on next page*

*Figure 2 continued*

**Figure supplement 2—source data 1.** Numerical data for *Figure 2—figure supplement 2*.

**Figure supplement 3.** The forces delivered using different stimulation probes.

**Figure supplement 3—source data 1.** Numerical data for *Figure 2—figure supplement 3*.

This is consistent with their known function as a larval gentle touch receptor (*Figure 2—figure supplement 1*; *Tsubouchi et al., 2012*; *Yan et al., 2013*).

The observation of different responses of c4da to the 30 and 60 µm probes suggests that c4da are more sensitive to smaller probes. To test this idea, we stimulated c4da using probes of different diameters (30, 60, 100, and 200 µm) at the distal regions (*Figure 2F*). We found that c4da made stronger responses ($\Delta F/F_0$) to smaller probes at the same force (*Figure 2G*). In addition, the slope of the force-$\Delta F/F_0$ curve increased as the probe became smaller (*Figure 2G*), reflecting an enhanced mechano-sensitivity to the change in stimulating force. To further understand this result, we calculated central pressure ($P_0$) and contact area ($A_c$) based on the 'sphere-surface' model (*Equations 4–7*, see **Materials** and **methods**), and then plotted the responses against these two parameters separately (*Figure 2H*). In the $P_0$-$\Delta F/F_0$ plots (left panel in *Figure 2H*), we found that increasing contact area decreased the threshold of cell activation but had only a minor impact on the slopes of the curves, suggesting that c4da have a robust sensitivity to the change in pressure. Similar observations were obtained when the pressures at other positions ($P_{x\ µm}$) were plotted against the responses ($\Delta F/F_0$) (*Figure 2—figure supplement 2*). In the $A_c$-$\Delta F/F_0$ plots (right panel in *Figure 2H*), when the contact area was the same, higher pressure led to stronger responses (*Figure 2H*), also consistent with c4da being a pressure sensor. The comparison of the $\Delta P$ (the change of pressure) and $\Delta A_c$ (the change of contact area) of different probes showed that for a given change of forces, the smaller probes caused a smaller $\Delta A_c$ but a greater $\Delta P$ in comparison to the larger probes (*Figure 2H*), explaining the higher sensitivity of c4da to the smaller probes.

To further understand how the sensory preference of c4da to localized poking contributes to the behavioral response of larvae in nociception, we performed the behavior assay of mechanical nociception using a modified Von Frey fiber. To test the effect of different probe sizes, we attached the glass probes (shorter than 1 mm) of different diameters (from 30 to 200 µm) to the end of a Von Frey fiber (*Figure 2I*). Using these modified fibers, we were able to deliver poking forces of 15–20 mN to fly larvae. Note that mean value and variation of the stimulating forces were independent on the probe size (*Figure 2—figure supplement 3*). The defensive rolling behaviors were recorded and scored to reflect the sensory response of c4da. We found that fly larvae showed stronger behavioral responses to smaller probes (*Figure 2J*), in agreement with our observations on the cellular level.

In summary, our results demonstrate that c4da are sensitive to mN-scale compressive forces applied onto their dendrites. In particular, c4da are more sensitive to smaller probes, consistent with c4da acting as a larval nociceptor to sense mechanical poking or sting from sharp objects, for example the wasp ovipositor (*Hwang et al., 2007*; *Robertson et al., 2013*; *Cerkvenik et al., 2017*).

## Morphological complexity likely contributes to the sensory features of c4da

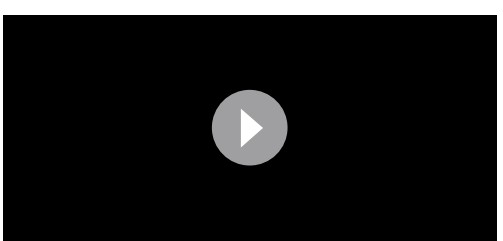

**Video 1.** The representative response of a c4da cell to the 4 mN force stimulus. The asterisk indicated the force application point and the white arrowhead indicated the soma.

https://elifesciences.org/articles/76574/figures#video1

Having characterized the mechanosensory responses of c4da, we wondered what could be the cellular mechanisms underlying the sensory preference of c4da to small probes. The key thing is to ensure that an adequate number of mechanosensory molecules can be activated upon a localized force. Intuitively, a densely distributed dendritic arbor would be helpful. We tested this hypothesis by studying the sensory responses of a knock-down mutant of *cut* ($ct^i$), a key transcription factor underlying the high morphological complexity of c4da (*Grueber et al., 2003*; *Jinushi-Nakao et al., 2007*).

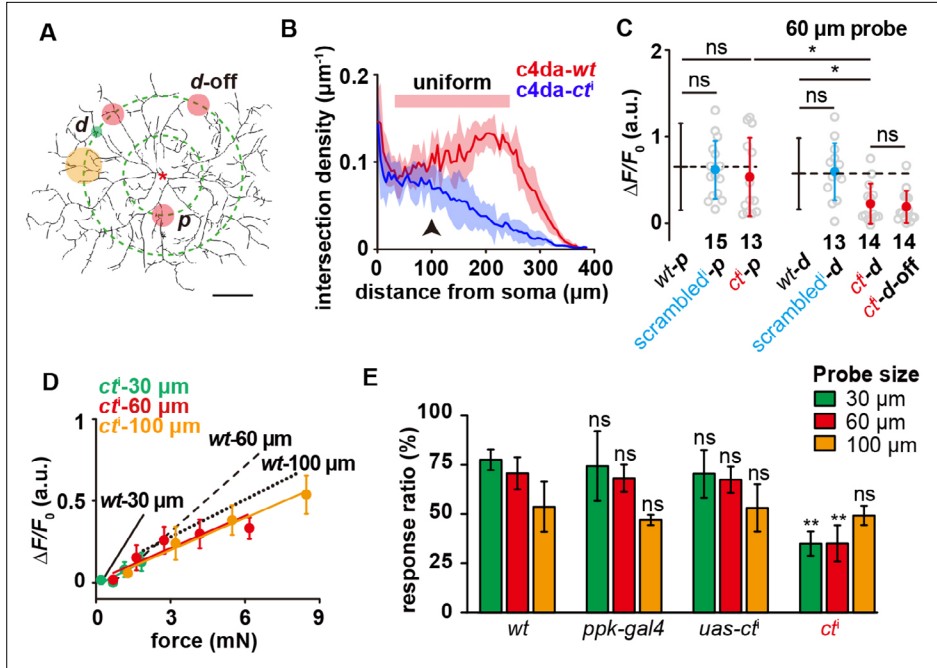

**Figure 3.** The contribution of dendritic morphology to the sensory features of c4da. (**A**) The schematic showing the force application points (filled circles, green: 30 μm probe, red: 60 μm probe, orange: 100 μm probe) on c4da-$ct^i$. The dashed concentric circles were 100 and 200 μm in radius, respectively. (**p**) proximal dendrite. (**d**) distal dendrite. d-off: the 'dendrite-off' region. Asterisk: the soma. Scale bar, 100 μm. Genotype: *ppk-gal4/+, ppk-cd4-tdtom/uas-ct*. (**B**) Modified Sholl analysis on the morphology of c4da-wt and c4da-$ct^i$. Note that there was a broad region in c4da-wt (red bar) in which the dendritic density was nearly constant. The shadow areas represented standard deviations. n=5 cells for each genotype. The black arrowhead indicated the regions of proximal dendrites. (**C**) The responses of c4da-$ct^i$ ($\Delta F/F_0$) to the force stimuli (4 mN) applied onto the proximal and distal dendrites using a 60 μm probe. The numbers of cells were indicated below the scattered data points. Mann Whitney U test was used. *: p<0.05. ns: no significance. $ct^i$: *ppk-gal4/20×uas-ivs-gcamp6s, ppk-cd4-tdtom/uas-ct*. *scrambled*$^i$: *ppk-gal4/20×uas-ivs-gcamp6s, ppk-cd4-tdtom/uas-scrambled*$^i$. (**D**) The responses of c4da-$ct^i$ ($\Delta F/F_0$) to the forces applied on the distal dendrites using the probes of different sizes. Data were presented as mean ± sem (n=10 cells). (**E**) The behavioral responses of $ct^i$ larvae to mechanical poking using the probes of different sizes. *wt: w1118. ppk-gal4: ppk-gal4; +/+. uas-ct*$^i$: *+/+; uas-ct*$^i$. *ct*$^i$: *ppk-gal4/+; uas-ct*$^i$/+. ppk-gal4* larvae: 30 μm probe (n=63 larvae from three experiments), 60 μm probe (n=60 larvae from three experiments) and 100 μm probe (n=68 larvae from three experiments). *uas-ct*$^i$ larvae: 30 μm probe (n=68 larvae from three experiments), 60 μm probe (n=72 larvae from three experiments) and 100 μm probe (n=63 larvae from three experiments). *ct*$^i$ larvae: 30 μm probe (n=97 larvae from four experiments), 60 μm probe (n=91 larvae from four experiments) and 100 μm probe (n=91 larvae from four experiments). One-way ANOVA test was used. **: p<0.01. ns: no significance. In panels (**C**), (**D**) and (**E**), the corresponding data from c4da-wt were provided for comparison. In panel (**C**) and (**E**), data were presented as mean ± std.

The online version of this article includes the following source data and figure supplement(s) for figure 3:

**Source data 1.** Numerical data for *Figure 3*.

**Figure supplement 1.** The expression and localization of the mechanosensory molecules and cytoskeletal elements in c4da-$ct^i$.

**Figure supplement 1—source data 1.** Numerical data for *Figure 3—figure supplement 1*.

---

C4da-$ct^i$ neurons showed a sparser dendritic morphology than c4da-wt (*Figure 3A*). Modified Sholl analysis showed that the dendritic density of c4da-wt was uniform across almost the entire dendritic field, while that of c4da-$ct^i$ was similar to c4da-wt in the proximal region but rapidly decreased towards the distal region (*Figure 3B*). In comparison to c4da-wt or c4da-scrambled RNAi, c4da-$ct^i$ showed a similar response to the stimuli on the proximal dendrites, but a significantly weaker response to those on the distal dendrites (*Figure 3C*). We noted that the overall responses of c4da-$ct^i$ to different probes were generally reduced and in particular, the responses to the 30 μm probe were reduced to the

greatest extent (*Figure 3D*). Consistent with the cellular observations, the defensive rolling behaviors of c4da-*ct*[i] in response to the mechanical poking of small probes (30 and 60 μm) were largely weakened in comparison to those of c4da-*wt*, while those to large probes (e.g. 100 μm) showed nearly no change (*Figure 3E*). Based on these observations, we conclude that the sensory preference of c4da to small probes was lost in c4da-*ct*[i].

Because Cut is a transcription factor, the reduction of its expression level might lead to other changes in addition to the disrupted morphology. To explore the effects of these potential factors, we performed several control experiments. First, we checked the expression level and subcellular localization of two mechanosensory molecules, that is Piezo and Ppk1/Ppk26, in c4da-*ct*[i]. They showed similar expression and localization as in c4da-*wt* (*Figure 3—figure supplement 1*). Second, the changes in dendritic cytoskeleton might indirectly affect mechanosensation by altering the mechanical homeostasis of the dendrites, so we checked the dendritic signals of F-actin and microtubules. No significant difference was found (*Figure 3—figure supplement 1*). These two observations suggest that although the reduction in the expression level of Cut decreases the number of dendritic branches, the expression and localization of mechanosensory molecules and cytoskeletal elements are nearly unchanged in the existing dendrites of c4da-*ct*[i]. Third, we also performed the same set of mechanical recording experiments on c3da-*wt*, in which there is no additional manipulation at the genetic level, but the dendritic density differs from that of c4da-*wt* in the way as in the case of c4da-*ct*[i] (*Figure 2—figure supplement 1*). The responses of c3da-*wt* to the distal stimuli were significantly weaker than those to the proximal stimuli (*Figure 2—figure supplement 1*). In addition, no preference to small probes was found in c3da-*wt* (*Figure 2—figure supplement 1*). These sensory features were markedly different from those in c4da-*wt*, but similar to those in c4da-*ct*[i]. In all, our observations are consistent with the dendritic morphology making a direct contribution in supporting the sensory features of c4da. Note that the potential contributions of other unknown factors cannot be absolutely excluded (see **Discussion**).

## Mechanosensitivity to lateral tension expands the force-receptive field

An unexpected finding in studying c4da-*ct*[i], primarily due to the sparse dendritic morphology, was that when the force probe was not directly placed on a dendrite but on an adjacent region without any dendrite (i.e. the 'dendrite-off' mode, denoted as '*d-off*' in the plots), the response of c4da was nearly unchanged, as if the force was directly applied on the dendrite (*Figure 3C*). We wondered if this reflects impalpable difference due to the overall reduction in the sensory response of c4da-*ct*[i] or that the dendrites of c4da have an expanded force-receptive field. To verify this, we performed the same experiments on c4da-*wt* (*d-off* in *Figure 2D*). The responses of c4da-*wt* to the *d-off* stimuli were also unchanged from those to the distal stimuli (*Figure 4A*). Further experiments on c4da-*wt* showed that the responses kept nearly constant until the force was applied at least 40–60 μm away from a dendrite (*Figure 4A*). Based on these results, we conclude that the dendrites of c4da have an expanded force-receptive field.

The expansion of force-receptive field suggests that in addition to the sensitivity to the pressure perpendicular to cuticle surface ($P_p$), c4da are also sensitive to lateral tension (parallel to cuticle surface, $T_L$). By simulating tissue mechanics of the larval fillet (*Figure 4—figure supplement 1*, also see **Materials** and **methods**), we calculated the distribution of $P_p$ and $T_L$ as a result of the poking stimuli from the 30 and 60 μm probes (indentation depth: 20 μm). In both cases, we noted that as the distance to the center of the force probe increased, $P_p$ showed a monotonic decrease but $T_L$ peaked at around 20–25 μm away from the probe center (*Figure 4B*). Therefore, modeling analysis predicts that if c4da were only sensitive to $\Delta P_p$, their sensory response to the *d-off* stimuli should decrease. However, if c4da were also sensitive to $\Delta T_L$, the increase in $T_L$ would compensate for the reduction in $P_p$. In this scenario, the effective force-receptive field is expanded.

Morphological analysis of c4da showed that almost all positions in the dendritic territory were within 20 μm distance from a dendrite (*Figure 4C*). To examine how the sensitivity to $T_L$ promotes the likelihood of cell activation, we overlaid the 2D profiles of $P_p$ and $T_L$ to the dendrites of c4da at random positions (*Figure 4C*) and calculated the probability of cell activation. Note that the calculations were made based on two assumptions. First, we assumed 10% of the peak value of $P_p$ or $T_L$ as the activation threshold of a dendritic segment. Second, we assumed a dendritic coverage threshold ($C_d$), i.e. the minimal amount of excited dendritic segments that would lead to neuronal excitation. To

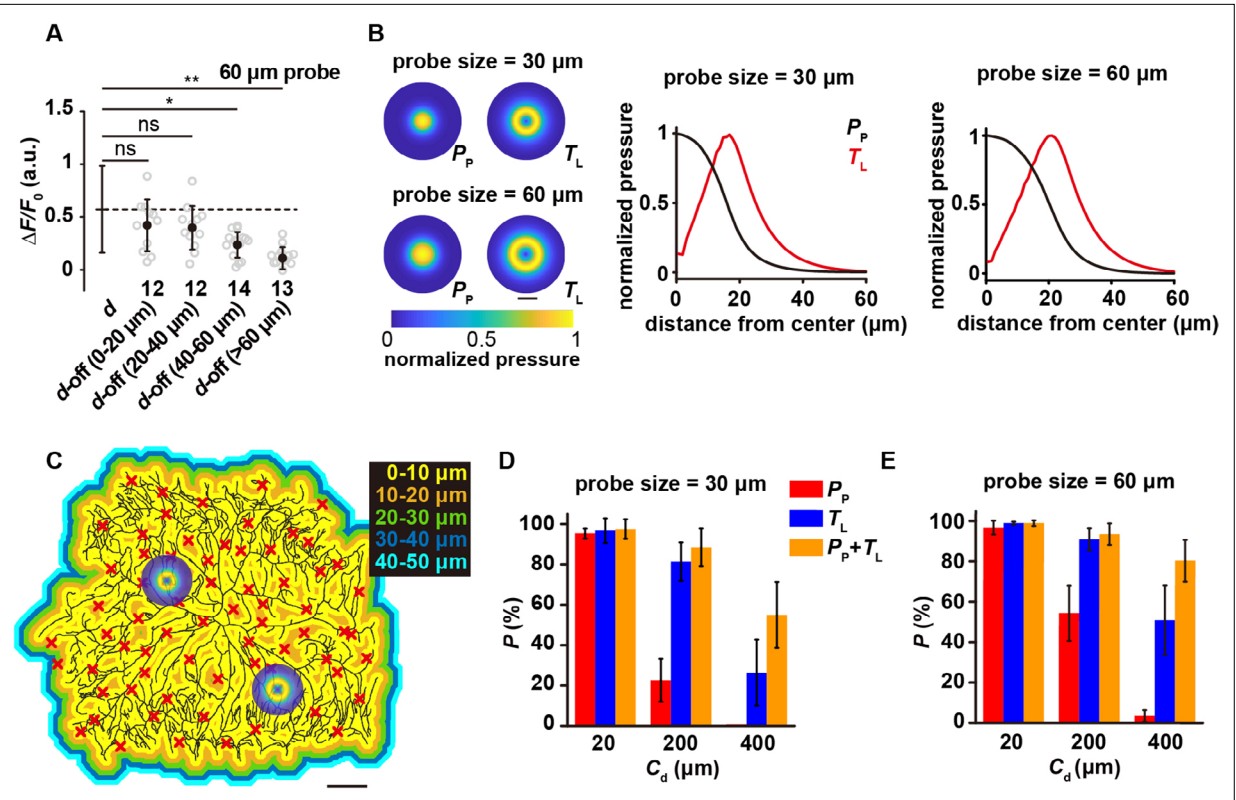

**Figure 4.** The mechanosensitivity of c4da to lateral tension. (**A**) The responses of c4da ($\Delta F/F_0$) to the *d-off* stimuli (4 mN, 60 μm probe) applied at different positions.The numbers of cells were indicated below the scattered data points. Mann Whitney U test was used. **: p<0.01. *: p<0.05. ns: no significance. Genotype: *ppk-gal4/+; ppk-cd4-tdtom/20×uas-ivs-gcamp6s*. (**B**) Left panel: representative heat maps showing the 2D distributions of pressure perpendicular to the cuticle surface ($P_P$) and tension parallel with the cuticle surface ($T_L$) of the 30 μm (upper) and 60 (lower) μm probes. Scale bar, 30 μm. Right panels: representative line profiles (normalized) showing the distribution of $P_P$ and $T_L$ versus the distance to the center of the force probe. Indentation depth: 20 μm. (**C**) The representative color map for the distance to the nearest dendrite, in which the distance was color-coded as indicated. Random positions were chosen (the red crosses) as the force application points in our simulations. Scale bar, 100 μm. (**D–E**) The activation probability in three conditions: (1) only sensitive to $P_P$; (2) only sensitive to $T_L$; (3) sensitive to both $P_P$ and $T_L$. The dendritic coverage threshold ($C_d$) was the minimal length of activated dendrites that could excite neuronal responses. For each condition, 100 random positions and two sizes of probes (panel **D**) 30 μm; (panel **E**) 60 μm were used in our simulations for each cell (n=5 cells). In panel (**A**), (**D**) and (**E**), data were presented as mean ± std.

The online version of this article includes the following source data and figure supplement(s) for figure 4:

**Source data 1.** Numerical data for *Figure 4*.

**Figure supplement 1.** Mechanical modeling analysis on the distribution of $P_P$ and $T_L$.

**Figure supplement 1—source data 1.** Numerical data for *Figure 4—figure supplement 1*.

explore the parameter space, we tested three values for $C_d$, i.e. 0.1% (20 μm) (low threshold), 1% (200 μm) (intermediate threshold) and 2% (400 μm) (high threshold) of the total dendritic length (mean ± std: 19560±1667 μm, n=5 cells). We considered three scenarios in our simulations, in which the cell is sensitive to either $P_P$ or $T_L$ or both. Simulation results showed that with the lower threshold assumption (e.g. $C_d$ = 20 μm), the cells could be activated in all three conditions. However, if the threshold was of intermediate (e.g. $C_d$ = 200 μm) or high (e.g. $C_d$ = 400 μm) level, lateral mechanosensitivity could largely enhance the probability of neuronal excitation (*Figure 4D–E*), consistent with our hypothesis. Therefore, our analysis suggests that the sensitivity to lateral tension enhances the sensitivity of c4da in force detection, especially when dendritic coverage is small.

## Piezo and Ppk1/Ppk26 differentially contribute to the mechanosensitivity of c4da

We then wondered what is the molecular basis underlying the mechanosensitivity of c4da, especially the sensitivity to lateral tension? Previous behavior assays suggest that Ppk1/Ppk26 and Piezo mediate

two parallel mechanosensory pathways in c4da (*Zhong et al., 2010*; *Kim et al., 2012*; *Gorczyca et al., 2014*; *Guo et al., 2014*; *Mauthner et al., 2014*). This remains to be confirmed at the cellular level. Moreover, it is also unclear how Ppk1/Ppk26 and Piezo contribute to the mechanosensitivity of c4da, respectively.

We first confirmed that the dendritic morphologies of c4da-*piezo*$^{KO}$, c4da-*ppk26*$^1$ (a null mutant) and c4da-*piezo*$^{KO}$, *ppk1*$^{Δ5}$ (a double null mutant) were not affected (*Figure 5A–B*), excluding the potential effect of morphological changes. We then measured the force-evoked responses of these mutants using two types of probes (30 and 60 μm) and at different dendritic regions. In all conditions, the responses of c4da-*ppk26*$^1$ were significantly reduced and the sensory preference to small probes was completely lost (*Figure 5C–D*). In contrast, the response of c4da-*piezo*$^{KO}$ was changed in a different way. C4da-*piezo*$^{KO}$ showed a mildly reduced response to the large probe (60 μm) but a significantly lower response to the small probe (30 μm). This phenotype was more prominent for the distal stimuli than for the proximal stimuli (*Figure 5C–D*). As a result, the preference to small probes was only attenuated for the proximal stimuli but completely lost for the distal stimuli (*Figure 5C–D*). Based on these results, we conclude that Ppk1/Ppk26 contributes to the overall mechanosensitivity of c4da and Piezo is particularly important for detecting localized forces. Furthermore, c4da-*piezo*$^{KO}$, *ppk1*$^{Δ5}$ (the double mutant) showed almost no response to any force stimuli (*Figure 5E*), suggesting that the contributions of Ppk1/Ppk26 and Piezo are additive. Finally, the responses of c4da-*piezo*$^{KO}$ and c4da-*ppk26*$^1$ to the **d-off** stimuli were comparable with those to the distal stimuli, showing that both mutants are still able to sense lateral tension (*Figure 5F*). However, their responses to the **d-off** stimuli were significantly weaker than that of c4da-*wt* (*Figure 5F*), suggesting that Ppk1/Ppk26 and Piezo both make contributions to the lateral mechanosensitivity of c4da.

To further verify the physiological contributions of Ppk1/Ppk26 and Piezo in mechanical nociception, we performed the behavior assay of mechanical nociception on these mutants using the 30 and 60 μm probes. C4da-*piezo*$^{KO}$ showed a largely reduced response to the 30 μm probe but an almost unchanged response to the 60 μm probe. Because the expression of Piezo is not restricted to c4da, we performed tissue-specific rescue experiment by driving the expression of Piezo using a c4da-specific driver (*ppk-gal4*). In the rescue strain, the behavioral defect of c4da-*piezo*$^{KO}$ in mechanical nociception could be rescued, suggesting that the behavior phenotype could be largely accounted for by the loss of Piezo in c4da (*Figure 5G*). Meanwhile, c4da-*ppk26*$^1$ showed very weak responses to the probes of both sizes and the double mutant (i.e. c4da-*piezo*$^{KO}$, *ppk1*$^{Δ5}$) showed almost no responses to all mechanical stimuli (*Figure 5G*). In all, these behavioral observations are consistent with the idea that Ppk1/Ppk26 and Piezo make differential but additive contributions in the c4da-mediated mechanical nociception.

## Ca-α1D contributes to the dendritic signal propagation in c4da

Finally, we addressed if the cellular mechanosensitivity of c4da relies on active signal propagation in the dendrites. The observation of the unchanged responses to the distal and proximal stimuli using the 60 μm probe (*Figure 2E*) supports the idea of active signal propagation in the dendrites. Meanwhile, the lower response to the distal stimuli than that to the proximal stimuli using the 30 μm probe in both c4da-*wt* (*Figure 2E*) and c4da-*piezo*$^{KO}$ (*Figure 5C–D*) suggests that the dendritic signal propagation mechanism is not all-or-none but graded, possibly dependent on the amounts of dendritic segments being activated. This hypothesis predicts that when the dendritic signal propagation, if any, is weakened, the cellular response of c4da to smaller probes would be affected to a greater extent. We then tested this hypothesis.

Previous studies showed that Ca-α1D, a voltage-gated calcium channel (VGCC), contributes to the somatic and dendritic calcium transient evoked by heat or laser stimulation of c4da (*Terada et al., 2016*; *Basak et al., 2021*), suggesting its function in facilitating dendritic signal propagation. Therefore, we first examined the roles of VGCCs in the force-evoked responses. When c4da-*wt* were stimulated by a 60 μm probe, the calcium increase was not only observed in the soma but also in the dendrites on the homolateral (p1 position: between the soma and the force probe; *Figure 6A*) and contralateral (p2 position: opposite side of the force probe; *Figure 6A*) sides of the stimulating forces. These observations demonstrate the occurrence of dendritic signal propagation. The calcium responses in both homolateral and contralateral dendrites can be suppressed by nimodipine (5 μM)

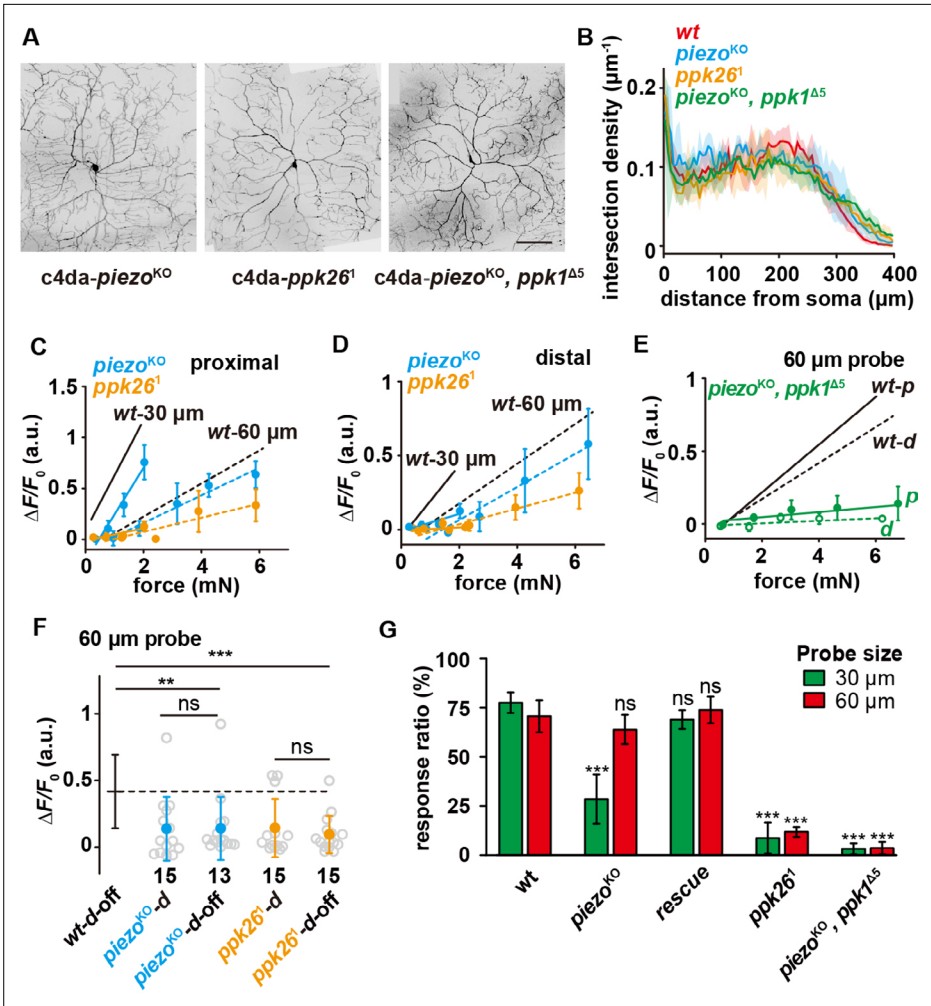

**Figure 5.** The differential contributions of Piezo and Ppk1/Ppk26 to the mechanosensitivity of c4da. (**A**) Representative images of c4da-*piezo*[KO] (*piezo*[KO]; *ppk-cd4-tdtom/+*), c4da-*ppk26*[1] (*uas-cd4-tdtom/ppk-gal4; ppk26*[1]), c4da-*piezo*[KO], *ppk*[Δ5] (*piezo*[KO], *ppk*[Δ5]; *ppk-cd4-tdtom/+*). Scale bar: 100 μm. (**B**) Modified Sholl analysis on the morphologies of c4da-*wt* (n=5), c4da-*piezo*[KO] (n=3), c4da-*ppk26*[1] (n=3) and c4da-*piezo*[KO], *ppk*[Δ5] (n=3). The shadow areas represented standard deviations. (**C–D**) The responses of c4da-*piezo*[KO] and c4da-*ppk26*[1] ($\Delta F/F_0$) to the proximal (**C**) and distal (**D**) stimuli delivered using the probes of two different sizes. Solid lines: 30 μm probe. Dashed lines: 60 μm probe. *piezo*[KO]: *piezo*[KO]; *ppk-gal4/20×uas-ivs-gcamp6s* (n=12 cells). *ppk26*[1]: *ppk-gal4/20×uas-ivs-gcamp6s; ppk26*[1] (n=12 cells). (**E**) The responses of c4da-*piezo*[KO], *ppk*[Δ5] ($\Delta F/F_0$) to the proximal and distal stimuli delivered using a 60 μm probe. Solid lines: proximal stimuli. Dashed lines: distal stimuli. Genotype: *piezo*[KO], *ppk1*[Δ5]: *piezo*[KO], *ppk1*[Δ5]; *ppk-gal4/20×uas-ivs-gcamp6s* (n=8 cells). (**F**) The responses of c4da-*piezo*[KO] and c4da-*ppk26*[1] ($\Delta F/F_0$) to the *d*-**off** stimuli (4 mN) delivered using a 60 μm probe. The numbers of cells were indicated below the scattered data points. Mann Whitney U test was used. \*\*: p<0.01. \*\*\*: p<0.001. ns: no significance. (**G**) The behavioral responses of the c4da-*piezo*[KO], c4da-*rescue*, c4da-*ppk26*[1] and c4da-*piezo*[KO], *ppk*[Δ5] larvae to mechanical poking using the probes of different sizes. *wt*: w1118. *piezo*[KO]: *piezo*[KO]; +/+. *rescue*: *piezo*[KO];*uas-gfp-piezo/ ppk-gal4*. *ppk26*[1]: +/+; *ppk26*[1]. *piezo*[KO], *ppk1*[Δ5]: *piezo*[KO], *ppk1*[Δ5]; +/+. *piezo*[KO] larvae: 30 μm probe (n=70 larvae from three experiments), 60 μm probe (n=63 larvae from three experiments). *rescue* larvae: 30 μm probe (n=50 larvae from three experiments), 60 μm probe (n=47 larvae from three experiments). *ppk26*[1] larvae: 30 μm probe (n=72 larvae from three experiments), 60 μm probe (n=93 larvae from three experiments). *piezo*[KO], *ppk1*[Δ5] larvae: 30 μm probe (n=61 larvae from three experiments), 60 μm probe (n=60 larvae from three experiments). One-way ANOVA test was used for the comparison among different genotypes. \*\*\*: p<0.001. ns: no significance. In panels (**C**), (**D**) and (**E**) data were presented as mean ± sem. In panels (**F**) and (**G**), data were presented as mean ± std. In panels (**C**), (**D**), (**E**), (**F**) and (**G**), the data from c4da-*wt* were provided for comparison.

The online version of this article includes the following source data for figure 5:

**Source data 1.** Numerical data for *Figure 5*.

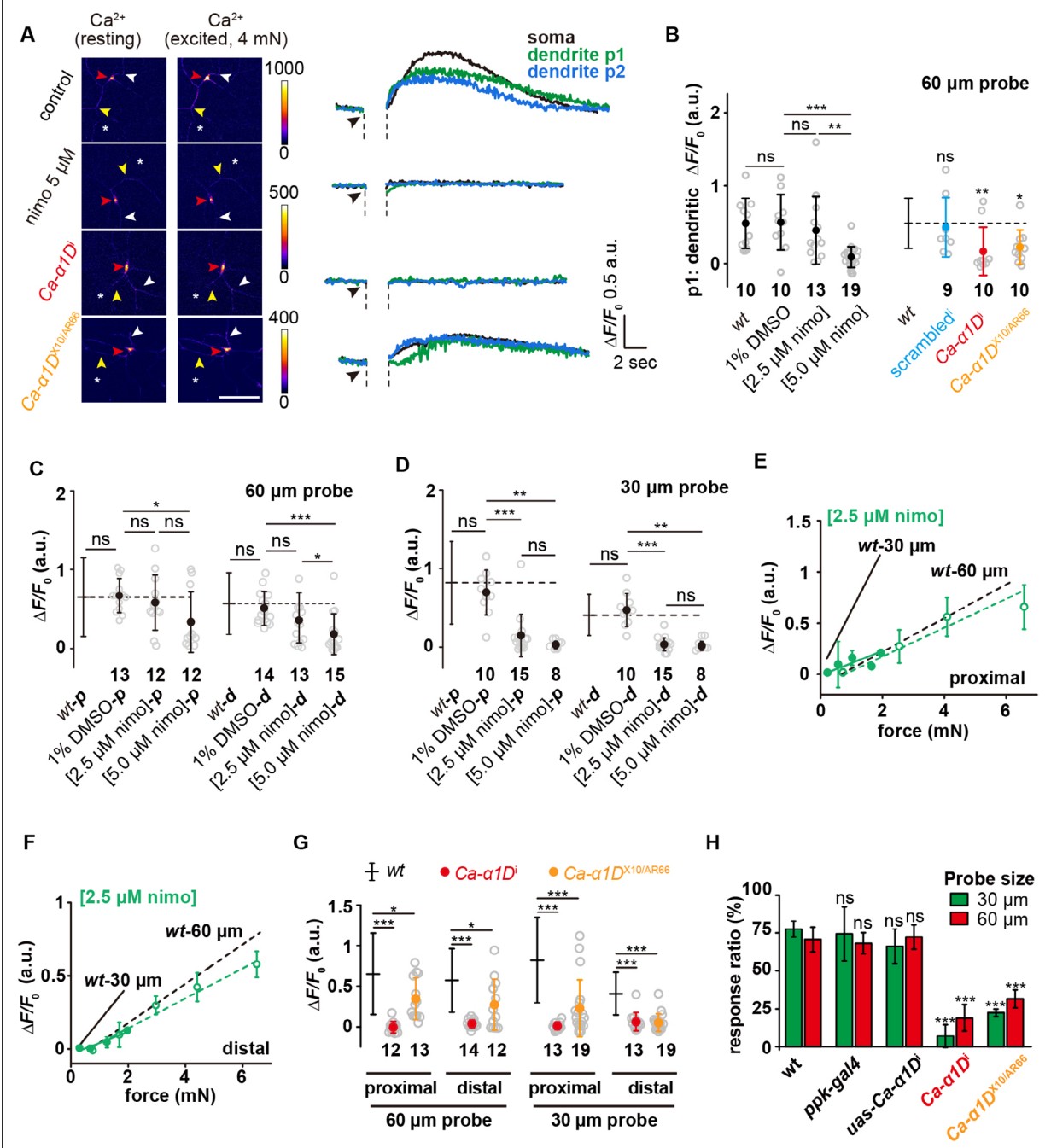

**Figure 6.** Active signal propagation in the dendrites of c4da.

(**A**) Representative images and curves showing the responses (ΔF/F₀) of the soma (red arrowhead), the homolateral dendrite (position p1: yellow arrowhead) and contralateral dendrite (position p2: white arrowhead) of c4da-*wt*, nimodipine-treated c4da-*wt*, c4da-*Ca-α1D*ⁱ and *Ca-α1D*^X10/AR66 to the stimuli (4 mN, 60 µm probe). The asterisk indicated the force application point. Scale bar: 50 µm. C4da-*wt*: *ppk-gal4/+; ppk-cd4-tdtom/20×uas-ivs-gcamp6s*. *Ca-α1D*ⁱ: *ppk-gal4/20×uas-ivs-gcamp6s; ppk-cd4-tdtom/ uas-Ca-α1D*ⁱ. *Ca-α1D*^X10/AR66: *Ca-α1D*^X10/ *Ca-α1D*^AR66; *ppk-gal4/20×uas-ivs-gcamp6s*. (**B**) Statistical quantification of calcium increases in the homolateral dendrites in response to the stimuli (4 mN, 60 µm probe). The genotypes were the same as those shown in panel (**A**). Mann Whitney U test was used for the comparison between c4da-*wt* and the mutants. Kruskal Wallis test was used for the comparison among the groups with drug treatment. ***: p<0.001. **: p<0.01. ns: no significance. (**C**) and (**D**) The responses (ΔF/F₀) of nimodipine treated c4da-*wt* to the proximal and distal stimuli from the probes of different sizes. (**C**): 60 µm probe stimuli. (**D**) : 30 µm probe stimuli. Kruskal Wallis test was used. ***: p<0.001. **: p<0.01. *: p<0.05. ns: no significance. (**E**) and (**F**) The responses (ΔF/F₀) of c4da-*wt* (n=10 cells) treated with 2.5 µM nimodipine to the proximal (**E**) and distal (**F**) stimuli delivered using the probes of two different sizes. Solid lines: 30 µm probe. Dashed lines: 60 µm probe. (**G**) The responses (ΔF/F₀) of c4da-*Ca-α1D*ⁱ and c4da-*Ca-α1D*^X10/AR66 to the stimuli applied at different dendritic regions (proximal and distal) and

*Figure 6 continued on next page*

*Figure 6 continued*

delivered using the probes of two different sizes (60 µm: 4 mN, 30 µm: 1.5 mN). (**H**) The behavioral responses of the c4da-Ca-α1D$^i$ larvae to mechanical poking using the probes of two different sizes. *wt: w1118. ppk-gal4: ppk-gal4; +/+. uas-Ca-α1D$^i$: +/+; uas-Ca-α1D$^i$. Ca-α1D$^i$: ppk-gal4/+; uas-Ca-α1D$^i$ /+. Ca-α1D$^{X10/AR66}$: Ca-α1D$^{X10}$/ Ca-α1D$^{AR66}$; +/+. uas-Ca-α1D$^i$* larvae: 30 µm probe (n=61 larvae from three experiments), 60 µm probe (n=61 larvae from three experiments). *Ca-α1D$^i$* larvae: 30 µm probe (n=56 larvae from three experiments), 60 µm probe (n=60 larvae from three experiments). *Ca-α1D$^{X10/AR66}$* larvae: 30 µm probe (n=43 larvae from three experiments), 60 µm probe (n=47 larvae from three experiments). One-way ANOVA test was used for the comparison among different genotypes. ***: p<0.001. ns: no significance. In panels (**B**), (**C**), (**D**), (**G**) and (**H**), data were presented as mean ± std. In panels (**B**), (**C**), (**D**) and (**G**), the numbers of cells were indicated below the scattered data points. In panels (**E**) and (**F**), data were presented as mean ± sem. In panels (**C**), (**D**), (**E**), (**F**), (**G**) and (**H**), the corresponding data from c4da-*wt* were provided for comparison.

The online version of this article includes the following source data and figure supplement(s) for figure 6:

**Source data 1.** Numerical data for *Figure 6*.

**Figure supplement 1.** Statistics quantification of the calcium increase in the contralateral dendrites in response to the force stimuli.

**Figure supplement 1—source data 1.** Numerical data for *Figure 6—figure supplement 1*.

**Figure supplement 2.** The expression and localization of Ca-α1D in vivo.

**Figure supplement 2—source data 1.** Gel image and numerical data for *Figure 6—figure supplement 2*.

(*Figure 6B* and *Figure 6—figure supplement 1*), an antagonist of VGCCs, suggesting that VGCCs contribute to the observed force-evoked dendritic calcium increases.

Drug titration experiments showed that in the condition of using a 60 µm probe, 5 µM nimodipine, the full inhibition concentration in the literatures (*Scriabine and van den Kerckhoff, 1988*; *Terada et al., 2016*), significantly suppressed the responses of c4da to both proximal and distal stimuli, while 2.5 µM nimodipine showed nearly no effect on the response to the proximal stimuli but a moderate suppression on that to the distal stimuli (*Figure 6C*). We interpreted this observation as when the VGCCs in the dendrites of c4da were partially inhibited, the responses to the distal stimuli were affected to a greater extent, suggesting that the dendritic signal propagation is more important in sensing distal stimuli. When a 30 µm probe was used, 2.5 µM nimodipine showed an inhibitory effect on the response to the proximal stimuli and an even stronger effect on that to the distal stimuli (*Figure 6D*). Further measurements on the responses of c4da to the full range of stimulating forces in the presence of 2.5 µM nimodipine showed that the drug largely suppressed the responses to the 30 µm probe but had only a mild effect on those to the 60 µm probe (*Figure 6E–F*), suggesting the loss of the sensory preference of c4da to small probes (*Figure 6E–F*). These observations demonstrate that the loss of VGCCs reduces the sensory preference to small probes, possibly by disrupting the dendritic signal propagation.

Inspired by the previous studies (*Eberl et al., 1998*; *Terada et al., 2016*), we then performed further experiments on the genomic (*Ca-α1D$^{X10/AR66}$*) and knockdown (*Ca-α1D$^i$*) mutants of *Ca-α1D*. *Ca-α1D$^{X10/AR66}$* has been genetically and functionally characterized in previous studies (*Eberl et al., 1998*; *Terada et al., 2016*). The *Ca-α1D$^i$* mutant has been used for functional studies in a previous report (*Terada et al., 2016*) and we here also demonstrated that the expression of *Ca-α1D$^i$* could significantly reduce the expression level of *Ca-α1D* (*Figure 6—figure supplement 2*). We noted that in comparison to c4da-*wt*, the calcium responses observed in the dendrites were largely reduced in these mutants (*Figure 6B* and *Figure 6—figure supplement 1*). In addition, the calcium responses in the soma of these mutants were also significantly reduced, in particular for the responses to the distal stimuli from the small probe (*Figure 6G*). Using the behavior assay, we found that the *Ca-α1D$^i$* and *Ca-α1D$^{X10/AR66}$* larvae showed largely weakened defensive behaviors in response to the mechanical poking, especially to those from the smaller probes (*Figure 6H*), which accords with our cellular observations. In all, our genetic and functional analysis suggests that Ca-α1D, as a subunit of VGCC, contributes to the sensory preference of c4da, likely by facilitating dendritic signal propagation. Further expression analysis using the existing sequencing datasets (*Li et al., 2022*) showed that Ca-α1D had a broad expression in various types of neurons, including those expressing *ppk1/ppk26* (e.g. c4da) (*Figure 6—figure supplement 2*). However, despite much effort, all of our further attempts to visualize the subcellular localization of Ca-α1D at both endogenous and over-expression conditions were so far not successful, possibly due to the low expression level as shown in the analysis on the sequencing datasets or other unknown technique reasons, for example that the tagged-Ca-α1D may

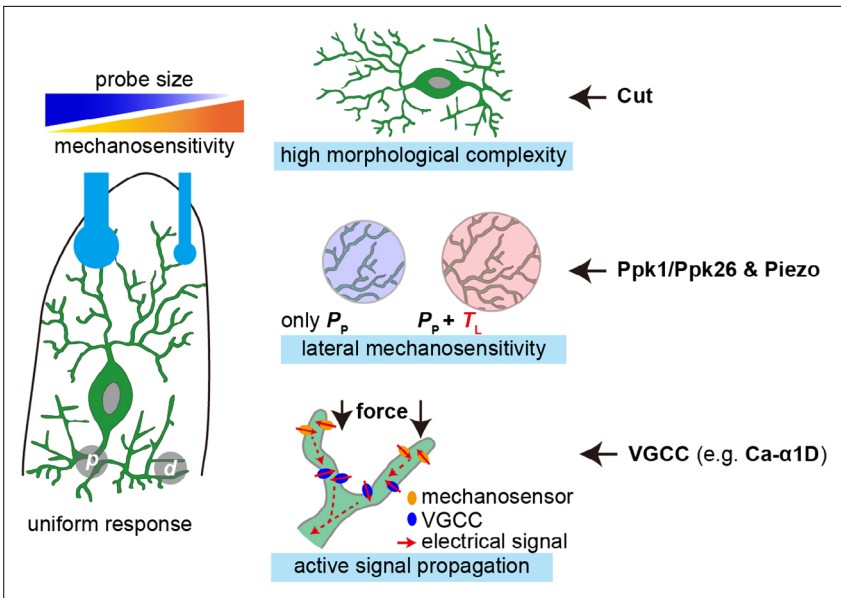

**Figure 7.** The cellular mechanisms that facilitate the mechanosensory features of c4da. Left panel: c4da showed a greater sensitivity to localized poking forces and made uniform responses to the forces applied at different dendritic regions (**p**) proximal stimuli. (**d**) distal stimuli. Right panel: the key cellular mechanisms that facilitate the sensory features of c4da and the important contributing molecules.

not be stable (*Figure 6—figure supplement 2*). Therefore, the cell biological features of Ca-α1D in c4da are still required to be characterized in future studies (see **Discussion**).

## Discussion

In the present study, we show that c4da are sensitive to mN-scale forces and could respond to the forces applied at different dendritic regions. In particular, c4da appear to be more sensitive to small probes, given that the applied forces are the same. We reveal three cellular mechanisms that could facilitate the sensory features of c4da (*Figure 7*). First, the high dendritic complexity ensures dense and uniform distribution of mechanosensory molecules. Second, the mechanosensitivity to lateral tension, which depends on both Ppk1/Ppk26 and Piezo, expands the force-receptive field of the dendrites. Third, the active signal propagation in the dendrites, possibly mediated by a voltage-gated calcium channel subunit Ca-α1D, promotes the overall sensitivity of c4da and has a more prominent effect on the sensory preference to small probes. We now discuss the potential implications of our findings in understanding the mechanosensory and nociceptive functions of c4da.

### Mechanical considerations on the differences between the semi-intact fillet preparation and intact larvae

In our assay, the semi-intact fillet preparation facilitates the coupling of mechanical stimulation to the recording of neuronal activities. However, it should be noted that the mechanics of our semi-intact fillet preparation is different from that of the intact animal, which is mainly reflected in two aspects. First, the fillet used in the present study is a tissue composite of cuticle, epidermis, neuron, extracellular matrix and muscle. In intact animals, this tissue composite is stretched and the tissular tension contributes to the mechanical homeostasis of c4da. In our semi-intact preparation, the tissue composite is also stretched when mounted on the PDMS pad. However, the resulting tissular tension is unlikely to be the same as that in the intact animals. This may change the mechanosensitivity of c4da. In addition, the change in the mechanical properties of tissue would alter the deformation of the fillet caused by the same force, which could also affect apparent force-dependent mechanosensitivity of c4da. Second, intact larvae are filled with the hemolymph but the fillet in our preparation is laid on a PDMS pad. The mechanical rigidity provided by the internal liquid pressure (from the hemolymph) and the muscle layer in the intact animals is possibly lower than that of the PDMS pad (*Kohane*

*et al., 2003*). Therefore, the same force may cause different tissue deformations and in turn excite the neurons to a different level. These differences may explain the apparent discrepancy between the lower force used in the present work (<5 mN) in comparison to the higher forces used in the behavioral assays in the previous reports (*Hwang et al., 2007*; *Zhong et al., 2010*; *Kim et al., 2012*; *Lopez-Bellido et al., 2019*). Although these mechanical differences may affect the range of forces required to excite c4da in different studies, we do not think that they would significantly affect the way of how c4da encode the spatial parameters of stimulating forces because the overall dendritic morphology and the expression of key sensory molecules are not affected.

## Implications for the mechano-nociceptive functions of c4da

C4da were first implicated as mechanoreceptor cells using the behavior assays (*Zhong et al., 2010*; *Kim et al., 2012*; *Gorczyca et al., 2014*; *Guo et al., 2014*; *Mauthner et al., 2014*). Cellular responses to force stimuli were observed on isolated *ppk1*-positive cells and whole mount larval fillet preparations (*Kim et al., 2012*; *Tsubouchi et al., 2012*; *Yan et al., 2013*; *Guo et al., 2014*; *Walcott et al., 2018*). Here, we provide quantitative characterizations on the responses of c4da to controlled forces. Our results add to the current model of understanding how c4da act as a mechanical nociceptor in the following two aspects.

First, at the cellular level, we find that c4da are more sensitive to smaller probes. This provides a cellular basis to understand the physiological function of c4da in sensing mechanical poking from sharp objects, for example the ovipositor of wasp (*Hwang et al., 2007*). From mechanical perspective, the geometry of the force probes (spherical in the present study) determines that for a given change of force, smaller probes caused a smaller $\Delta A_c$ but a greater $\Delta P$ in comparison to larger probes. Because c4da are pressure sensors, the greater $\Delta P$ would elicit a greater change in the neuronal response. Meanwhile, c4da need to ensure that the $\Delta P$ over a small contact interface could be detected, integrated and amplified in order to excite neuronal responses. For this purpose, c4da develop several optimization mechanisms at the cellular level (*Figure 7*). First, complex dendritic morphology allows small probes to have a high dendritic coverage. Second, lateral mechanosensitivity expands the force-receptive field of dendrites, thereby acting as a mechanical amplifier. Third, active signal propagation, as an intracellular electrochemical amplifier, could compensate for signal attenuation in the dendrites. Altogether, these mechanisms optimize the mechanosensitivity of c4da by increasing the probability of activating the mechanosensory pathways that are required to excite neuronal responses.

An recent study reported that c4da can be excited by focal laser ablation through a noncontact pathway, namely that when the laser stimuli were not directly applied onto the dendrites, the neuronal responses could only be observed in axon but not in dendrites (*Basak et al., 2021*). This pathway is thought be mediated by extracellular molecules released from damaged cells and implies that at least in some signaling processes of c4da, the axon has a lower threshold of activation than the dendrites (*Basak et al., 2021*). In the present work, we did not observe noncontact responses, including those using the **d-off** stimuli. We think that there are mainly two reasons for this difference. First, in our experiments, the small deformation constraint minimizes tissue damage (*Figure 1—figure supplement 1*). Second, in our semi-intact preparation, the tissue was bathed in the insect hemolymph-like buffer, so the extracellular signaling molecules, if any, might be rapidly diluted and are unlikely to reach the effective working concentration. Therefore, we argue that mechanical stimulations used in our study primarily stimulate the contact pathway, in which the dendrites are activated in the first place. This is also supported by two additional observations. First, we note that the force-evoked neuronal responses are dependent on Ppk1/Ppk26 and Piezo channels. These channels have clear dendritic localizations and are expected to contribute to the dendritic responses. Second, although the dendritic localization of Ca-α1D is still required to be clarified, the dependence of dendritic responses on VGCCs (*Figure 6B*) suggests that an active signal propagation mechanism is involved in the force-evoked dendritic responses. Comparatively speaking, we think that the previous study provides novel insights into how c4da respond to local tissue damage (*Basak et al., 2021*), while our study advances the model of how c4da detect forces as a mechanoreceptor cell. These two studies, in a complementary way, promote our understanding of the function and working mechanism of c4da as a nociceptor.

Second, at the molecular level, we find that Piezo and Ppk1/Ppk26 make differential contributions to the mechanosensitivity of c4da. Previous behavior assays suggest that Ppk1/Ppk26 and Piezo in

c4da mediate two parallel pathways in mechanical nociception (*Kim et al., 2012*; *Gorczyca et al., 2014*; *Guo et al., 2014*; *Mauthner et al., 2014*). This raises the question of why c4da need two sets of mechanosensory pathways. Furthermore, patch clamp recordings showed that the force-evoked electrical response of isolated *ppk*-positive cells is entirely dependent on Piezo (*Kim et al., 2012*), which contradicts the result from the behavior assay. Therefore, it is necessary to explore how Ppk1/Ppk26 and Piezo contribute to the in-vivo mechanosensitivity of c4da. Here, we discover that Piezo is particularly important for c4da to detect localized forces but plays a relatively minor role in sensing large probes. In contrast, the loss of Ppk1/Ppk26 reduces the responses of c4da to the entire range of stimulating forces used in our experiments, suggesting the contribution of Ppk1/Ppk26 to the overall mechanosensitivity. In addition, the lack of responses in the double knockout mutant strain confirms that the contributions of Ppk1/Ppk26 and Piezo are additive, as suggested by the previous behavior assays (*Kim et al., 2012*). Therefore, our findings add to the current model by showing the differential roles of Ppk1/Ppk26 and Piezo in mechanosensation, and in particular, Piezo seems to act as a functional module to specifically enhance the mechanosensory features of c4da. Furthermore, we also show that Ppk1/Ppk26 and Piezo both contribute to the sensitivity of c4da to lateral tension. Consistent with this idea, members in the Piezo and the DEG/ENaC channel families have been proposed to be gated by the changes in lateral membrane tension (*Cueva et al., 2007*; *Guo and MacKinnon, 2017*; *Liang and Howard, 2018*; *Lin et al., 2019*). However, it remains unclear how Piezo could facilitate the greater sensitivity to localized forces. We think that the underlying molecular mechanisms may reside in the gating property of Piezo, such as gating sensitivity, conductance and the property of the gating force.

## Contribution of dendritic morphology to the function of c4da

Based on the observations in the $ct^j$ strain (*Figure 3*), we argue that the morphological complexity is a key factor in supporting the sensory features of c4da. Because Cut is a transcription factor that contributes to the morphological determination of c4da, it is formally possible that the altered mechanosensory responses observed in the $ct^j$ mutant is due to other downstream changes. While this possibility cannot be absolutely excluded, we think that the contribution of dendritic morphology is likely valid based on several lines of evidence and thoughts. First, the responses of c4da-$ct^j$ to the stimuli on the proximal dendritic region, where the dendritic density is similar to that of c4da-*wt*, is comparable to those of c4da-*wt*, suggesting that the essential mechanosensory elements are fairly normal in the existing dendrites of c4da-$ct^j$ (*Figure 3C*). Second, the expression and localization of mechanosensory molecules (Piezo and Ppk1/Ppk26) and cytoskeletal elements (F-actin and microtubule) in c4da-$ct^j$ are comparable to those in c4da-*wt* (*Figure 3—figure supplement 1*), although it is possible that the fine organization of these molecular components could still be different in c4da-$ct^j$. Third, at least in terms of how a cell responds to different probe sizes and to the stimuli at different dendritic regions, c4da-$ct^j$ are similar to c3da-*wt*, whose morphological pattern is markedly different from c4da-*wt* but close to c4da-$ct^j$ (*Figure 2—figure supplement 1*). Assuming that the spatial parameters of forces are primarily encoded at the cellular level, the similarity between c3da-*wt* and c4da-$ct^j$ would reflect the contribution of dendritic morphology. Therefore, our observations are consistent with the idea that the dendritic complexity is a contributing parameter underlying the force-encoding mechanism of c4da.

## Materials and methods
### Fly stocks

All flies were maintained on standard medium at 23–25°C in 12:12 light-dark cycles. *20×uas-ivs-gcamp6s* (BL42746, BL42749), *ppk-gal4* (BL32078, BL32079), *uas-cd4-tdTom* (BL35837, BL35841), *ppk-cd4-tdtom* (BL35845), *ppk-cd4-tdgfp* (BL35842, BL35843) *uas-lifeact-rfp* (BL58715), *Ca-α1D*[X10] (BL25141) and *Mi{ET1}Ca-α1D*[MB06807] (BL25527) were from Bloomington *Drosophila* Stock Center (BDSC). *Gal4-19-12* was from the Jan Lab (UCSF). *ppk26-gfp* was from the Tracey Lab (Indiana University). *Uas-mcherry-jupiter* was from the Han Lab (Cornell University). *Tmc-gal4*, *gfp-piezo*, *piezo*[KO], *uas-gfp-piezo*, *ppk26*[1] and *piezo*[KO], *ppk*[Δ5] and were from Wei Zhang (Tsinghua University). *tub-gal4* was from Bing Zhou (Tsinghua University). *ct*[j] (THU1309), *Ca-α1D*[j] (THU0766) and the control strains were from Tsinghua Fly Center. *Ca-α1D*[AR66] was from Daniel Eberl (University of Iowa).

## Larval fillet preparation

Third instar larvae were dissected on a 35 mm dish coated with a PDMS pad (thickness: 1 mm) in the insect hemolymph-like (HL) buffer (*Stewart et al., 1994*). The cuticle, epidermis and muscle tissue were kept as intact as possible. The fillet was mounted on the PDMS pad using 8–10 insect pins (Austerlitz Insect Pins, Czech) and kept as stretched. HL buffer: 103 mM NaCl (10019318 Sinopharm Chemical Reagent Co., Ltd., Shanghai, China), 3 mM KCl (P9541 Sigma-Aldrich, USA), 5 mM TES (T5691 Sigma-Aldrich, USA), 8 mM trehalose (T1067 Sigma-Aldrich, USA), 10 mM glucose (G7528 Sigma-Aldrich, USA), 5 mM sucrose (V900116 Sigma-Aldrich, USA), 26 mM NaHCO$_3$ (A500873 Sangon Biotech, Co., Lt., Shanghai, China), 1 mM NaH$_2$PO$_4$ (S8282 Sigma-Aldrich, USA), 2 mM CaCl$_2$ (Xilong Scientific Co., Ltd., Shantou, China), and 4 mM MgCl$_2$ (M8266 Sigma-Aldrich, USA). The PDMS pad was made using the Sylgard 184 kit (Dow, UAS).

## Mechanical device setup and calibration

The mechanical device consisted of a piezo actuator (PZT 150/7/60 vs 12, SuZhou Micro Automation Technology Co., Ltd., Suzhou, China), a strain gauge (customized in Nanjing Bio-inspired-tech Co., Ltd., Nanjing, China) and a force probe. The stepping distance (*D*) of the piezo actuator and the deflection of the metal beam (*B*) were measured using a digital camera with a pixel size of 1.1 µm (AO-3M630, AOSVI optical instrument Co., Ltd. Shenzhen, China). The voltage readout of the strain gauge after the addition of pure water with a minimal step of 100 µL (i.e. 0.98 mN) was recorded using a data acquisition card (NBIT-DSU-2404A, Nanjing Bio-inspired-tech Co., Ltd., Nanjing, China). The force probes were made from capillary glass tubes using an electrode puller (PC-10, Narishige, Japan) and then polished using a microforge (MF-830, Narishige, Japan). Finally, the assembled mechanical device was mounted on the working stage of the spinning-disk confocal microscope (Andor, UK).

## Confocal microscopy

The calcium fluorescent signals were recorded using an Andor inverted spinning disk confocal system (Andor, UK) equipped with an inverted microscope (Olympus, IX73), an iXon 897 EMCCD and a long working-distance 20×objective (UCPlanFL N, N.A. 0.70, working distance: 2.1 mm) (Olympus, Japan).

The images of Piezo-GFP, Ppk26-GFP, mCherry-Jupiter and Lifeact-RFP were acquired using a Zeiss 780 confocal microscope equipped with a 63×objective (Zeiss Plan-Apochromat, 1.4 N.A., Germany) and GaAsP detectors (Zeiss, Germany).

## Scanning electron microscopy

The scanning electron microscopy (SEM) pictures of the glass probes were imaged using FEI Quanta 200 (Thermofisher, USA) with 15-kV voltages and 500×magnification in the high vacuum mode.

## Modified behavior assay for mechanical nociception

The Von Frey fiber (6lb. Omniflex Line, Zebco) was mounted onto a holder. The free fiber was 20 mm long. The customized glass probe was fixed at the free end of the fiber using ergo 5400 (Kisling, Switzerland). The poking force was measured using an electronic precision balance (XPR204S, Mettler Toledo, USA) and was found to be independent on probe size (*Figure 2—figure supplement 3*). The behavior assays were carried out as the previously described (*Hwang et al., 2007*; *Zhong et al., 2010*). Briefly, we used the modified fiber to poke the dorsal side of larval middle segments and recorded their behavioral responses. If the larva showed a typical rolling behavior (rolling over 360°) upon each mechanical stimulation, it was considered as making an effective response. The percentage of responding animals was then calculated. Statistical analysis was performed using one-way ANOVA using Matlab (MathWorks, USA).

## Drug treatment

Nimodipine (N149 Sigma-Aldrich, USA) was dissolved in DMSO (276855 Sigma-Aldrich, USA). The incubation time for all drugs was 20 min.

## The 'sphere-surface' contact mechanics model

Based on the classical 'sphere-surface' contact model (*Johnson, 1985*), we had

$$f = \frac{4}{3} E^* r^{\frac{1}{2}} d^{\frac{3}{2}} \tag{1}$$

$$E^* \approx \frac{E_{\text{PDMS}}}{1-\nu^2} \tag{2}$$

where $f$ was the total force, $r$ was the radius of the sphere, $E_{\text{PDMS}}$ was the elastic modulus of PDMS, $E^*$ was the effective modulus, $\nu$ was the Poisson's ratio of PDMS and $d$ was the indentation depth. $d$ can be calculated as

$$d = D - B \tag{3}$$

where $D$ was the stepping displacement generated by the piezo stack actuator and $B$ was the bending deflection of the metal beam (*Figure 1A*). In this model, the pressure at the center of contact area ($P_0$) can be calculated as

$$P_0 = \frac{3f}{2\pi a^2} \tag{4}$$

where $a$ was the radius of the contact area and could be calculated as

$$a = \sqrt{rd} \tag{5}$$

The area of contact surface ($A_c$) can be calculated as

$$A_c = 2\pi r \left( r - \sqrt{r^2 - rd} \right) \tag{6}$$

The pressures at a given point to the soma can be calculated as

$$P_{x\,\mu m} = P_0 (1 - \frac{x^2}{a^2})^{\frac{1}{2}} \tag{7}$$

where $x$ (μm) was the distance to the center of contact area.

## Finite element analysis

To theoretically investigate the stress field in the cuticle caused by indentation, finite element simulations were performed using the commercial software ABAQUS 6.14.1. (Dassault Systèmes, France). Due to symmetry of the load and the geometry, an axisymmetric mechanical model was established (*Figure 4—figure supplement 1*). The model includes the spherical probe and the underlying composite that consists of two layers of materials: (1) i.e. the cuticle layer (thickness 10 μm, elastic modulus $E_{\text{cuticle}}$ = 1–10 MPa, Poisson's ratio $\nu_{\text{cuticle}}$=0.45); (2) the underlying PDMS substrate (thickness 1000 μm, elastic modulus $E_{\text{PDMS}}$ = 2.6 MPa, Poisson's ratio $\nu_{\text{PDMS}}$=0.45). Because the muscle layer (elastic modulus $E_{\text{muscle}}$ = 10 kPa) was much more compliant than cuticle and PDMS (*Kot et al., 2012*), it is expected to make little contribution to the force distribution. Therefore, we omitted the muscle layer in our finite element model for simplicity. The probe was treated as a rigid object, while the cuticle layer and PDMS substrate were assumed to be linearly elastic. Displacement load in the vertical direction was applied to the probe. The lower surface of the cuticle layer and the upper surface of the PDMS substrate were assumed to be tightly coupled. Fixed boundary conditions were applied to the bottom surface of the PDMS substrate. The compressive pressure (perpendicular to cuticle surface) and lateral (parallel to cuticle surface) tension were both calculated. The simulation of random positioned stimuli was performed using Matlab.

## Image analysis

GCaMP6s signal was measured using Fiji (*Schindelin et al., 2012*). The calibration bars and scale bars were generated in Fiji. Sholl analysis on neuronal morphology was carried out using Fiji (*Ferreira et al., 2014*). The density of intersections was calculated as the number of intersections divided by the circumference of corresponding circle in the Sholl analysis (*Figure 3*, *Figure 5* and *Figure 2—figure supplement 1*).

## The single-cell transcriptomic atlas analysis

We used the FlyCellAtlas (*Li et al., 2022*) single-cell RNA-sequencing datasets to explore the expression profiles of *Ca-a1D*. Specifically, we downloaded the stringent analysis result of the Body

10×Genomic sample from FlyCellAtlas website. We visualized the expression pattern with the use of Python packages scanpy (version 1.9.1) (*Wolf et al., 2018*) and anndata (version 0.8.0) (*Virshup et al., 2021*). We first explored the expression pattern of *Ca-a1D* in the whole dataset. To further check if the *ppk*-specific neurons also express *Ca-a1D*, we plotted the expression of both genes in the neuron clusters.

## Conventional data plotting and statistical analysis

Data plotting and statistical analysis were performed using Origin (OriginLab, USA). The heat maps of the $P_\text{P}$ and $T_\text{L}$ were generated using Matlab. Statistical analysis was performed using Origin.

## Generation of transgenic flies for Ca-α1D expression

To construct mCherry tagged Ca-α1D expressed under its own promoter, the coding sequence of *Ca-α1D*-RK transcript was fused with mCherry tag and its promoter sequence (a 3 kb sequence upstream of the *Ca-α1D*-RK coding region). The regulation sequence at 3' end, which is ~2.3 kb downstream of the Ca-α1D-RK coding region, was fused at the 3' end. The entire construct of 'promoter-mCherry-*Ca-α1D* –3' regulation sequence' was then cloned into pJFRC81 (Addgene 36432) between the HindIII and FseI sites. The plasmid was then injected into the *attp40* strain to generate the promoter fusion mCherry-Ca-α1D transgenic line.

To construct *uas-mcherry-Ca-α1D*, the coding sequence of mcherry-*Ca-α1D*-RK was cloned into pJFRC81 between the NotI and XbaI sites. The plasmid was then injected into the *attp2* strain to generate the *uas-mcherry-Ca-α1D* transgenic fly.

## Immunostaining

Immunostaining followed the procedure described previously (*Hsu et al., 2020*). 3rd larvae were filleted in PBS and fixed in 4% PFA in PBS. The primary antibody used was: chicken anti-Dmca1D (1:100, shared by John Y. Kuwada). Secondary antibody used was: Donkey anti-chicken Alexa Fluor 488 (1:500, SIGMA, 18C0824).

## RT-PCR

The total RNA was extracted from whole larva using the RNeasy Mini Kit (QIAGEN, 74104). The cDNA was synthesized using SuperScript III Reverse Transcriptase (RT) (Invitrogen, 18080051). All polymerase chain reactions were performed using the Phusion DNA polymerase (New England Biolabs, Ipswich, MA). *Rp49* was used as the reference gene. *Rp49* forward primer: 5'-TACAGGCCCAAGATCGTGAA-3'. *Rp49* reverse primer: 5'-TCTCCTTGCGCTTCTTGGA-3'. *Ca-α1D* forward primer: 5'-CATTGCAAACATTCCGGAAACC-3'. *Ca-α1D* reverse primer: 5'-CAATTGGGAGAGTGCAGTACT-3'. The gel image was taken using the Tanon image system (SHTN 1600, China). The intensity of DNA bands on the gel images was analyzed using Fiji.

## Acknowledgements

The authors thank Wei Zhang (Tsinghua University), Bing Zhou (Tsinghua University), Dan Tracey (Indiana University), Chun Han (Cornell University), Daniel Eberl (University of Iowa) and the Jan lab (UCSF) for sharing fly strains. We thank John Y Kuwada (University of Michigan) for sharing the antibody. Special thanks to the electron microscopy facility and fly facility in Tsinghua University. We acknowledge our funding from National Natural Sciences Foundation of China (31922018, 32070704), Tsinghua-Peking Center for Life Sciences, Beijing Advanced Innovation Center for Structural Biology and IDG/McGovern Institute for Brain Research (Tsinghua University).

# Additional information

## Funding

| Funder | Grant reference number | Author |
|---|---|---|
| National Natural Science Foundation of China | 31922018 | Xin Liang |
| National Natural Science Foundation of China | 32070704 | Xin Liang |

The funders had no role in study design, data collection and interpretation, or the decision to submit the work for publication.

## Author contributions

Zhen Liu, Conceptualization, Investigation, Methodology, Writing - original draft, Writing - review and editing; Meng-Hua Wu, Qi-Xuan Wang, Shao-Zhen Lin, Investigation; Xi-Qiao Feng, Supervision, Methodology; Bo Li, Supervision, Investigation, Methodology, Writing - review and editing; Xin Liang, Conceptualization, Formal analysis, Supervision, Funding acquisition, Investigation, Methodology, Writing - original draft, Project administration, Writing - review and editing

## Author ORCIDs

Xin Liang http://orcid.org/0000-0001-7915-8094

## Decision letter and Author response

Decision letter https://doi.org/10.7554/eLife.76574.sa1
Author response https://doi.org/10.7554/eLife.76574.sa2

# Additional files

## Supplementary files

• Transparent reporting form

## Data availability

All data generated or analysed during this study are included in the manuscript and supporting files. The source data for all plots have been provided as Excel files.

The following previously published datasets were used:

| Author(s) | Year | Dataset title | Dataset URL | Database and Identifier |
|---|---|---|---|---|
| Li H | 2021 | Fly Cell Atlas: single-cell transcriptomes of the entire adult *Drosophila* - Smartseq2 dataset | https://www.ebi.ac.uk/arrayexpress/experiments/E-MTAB-10628/ | ArrayExpress, E-MTAB-10628 |
| Maxime DW, Li H, Janssens J | 2021 | The Fly Cell Atlas: single-cell transcriptomes of the entire adult *Drosophila* - 10x dataset | https://www.ebi.ac.uk/arrayexpress/experiments/E-MTAB-10519/ | ArrayExpress, E-MTAB-10519 |

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
