## [Editor Report]

Liu et al. present a fascinating study that significantly advances fundamental knowledge about the molecular and cellular pathways underlying mechanical nociception. The use of a combination of fine biophysics and neurogenetics provides unprecedented insight into mechanosensory functions in an intact tissue environment of the *Drosophila* larva. The results of this work have strong implications for our understanding of the sensation of acute pain.

---

## [Decision Letter]

**Decision letter after peer review:**

Thank you for submitting your article "*Drosophila* Mechanical Nociceptors Preferentially Sense Localized Poking" for consideration by *eLife*. Your article has been reviewed by 3 peer reviewers, and the evaluation has been overseen by a Reviewing Editor and Claude Desplan as the Senior Editor. The following individuals involved in review of your submission have agreed to reveal their identity: Martin Göpfert (Reviewer #3).

The reviewers have discussed at length their reviews with one another, and the Reviewing Editor has drafted this to help you prepare a revised submission. The reviewers were generally very impressed by the quality of your study and the novelty of your results. Nonetheless, a series of significant technical concerns were raised. We are confident that you will be able to address these concerns, and ask that you do so in a revised manuscript. Although the proposed revisions will entail additional work and new data, we encourage you to build on existing results. Specially:

1. Please check if dendrite/epidermal breakage occurs with the used probes. Ruling out such breakages is important to evaluate the data and its interpretation both at the level of the calcium signals and behavior. You should use ppk-tdtomato or ppk-mCD8GFP for this to make sure dendrites are bright enough and visible. Also, magnification higher than 20X is probably needed.

2. Please use the existing set of recordings to evaluate the possibility of dendritic calcium signals. The idea is to analyze the spatial distribution of dendritic calcium responses to the pokes. If no or very weak dendritic calcium signals are found, this observation should be stated in the manuscript. If significant dendritic calcium signals are observed, you should determine the dependence of these signals on Ca-alpha1D.

3. Please validate the RNAi knockdown and mutant results, particularly for Ca-alpha1D and piezo. For the RNAi lines used, qPCR data on the knockdown-efficiency should be added. If the qPCR do not work, you should consider using piezo-GFP as a readout for assessing the knockdown efficiency. For Ca-alpha1D, you should check the relevant mutants that could be used to provide better genetic evidence in support of a role for Ca-alpha1D. Data on protein localization is presented for the mechanosensitive channels, but not for Ca-alpha1D. Knowing the distribution of the Ca-alpha1D subunit is critical to the interpretation of the rest of your data. To do this, we recommend you use an antibody against Ca-alpha1D that was generated in I-Uen Hsu et al., PNAS 2020, PMID:33168737. Confirming expression, localization and RNAi knockdown should be feasible if the antibody proves to be specific enough. If the antibody's specificity is insufficient, you should check whether Ca-alpha1D is expressed in the cell using e.g. Mi{ET1}Ca-α1D[MB06807] that is available at Bloomington. For mutant alleles of Ca-alpha1D, combinations of available reagents were used in a previous study cited by the author (Terada et al. *ELife* 2016): Ca-α1DX7/AR66 and Ca-α1DX10/AR66. While we realize these extra experiments represent substantial work, their results will be important to clarify the role of Ca-alpha1D and piezo to support several or your statements. In case the experiments listed above are unsuccessful, we ask you to tone down the interpretation of your results, which would be unfortunate.

4. Please check/redo the calculations related the elastic modulus (PDMS vs. larval cuticle) and clarify the differences in the mechanical properties of the open filet preparation vs. the intact animal. In particular, transferring the elastic modulus of muscles characterized in humans to flies does not seem to be legitimate. The modeling should be repeated with a realistic value for the elastic modulus of the larva, along with an improved discussion of how the force distributions would compare to those in an intact larva. In the discussion, you should address the apparent discrepancy between the force used in your work (<5 mN) compared to the 5-10x higher forces typically used in previous publications studying rolling responses in live larvae.

5. Please provide a minimal set of controls for behavioral data. This should include the driver control for the RNAi, the driver alone, and the driver plus RNAi. You have conducted these controls for the experiments related to cut but you are missing driver alone for Ca-alpha1D.

6. Please address the points raised by the reviewers regarding the interpretation of the results, the statistics and grammatical errors. Among the missing references, it would be important to incorporate into the discussion the following paper from the Howard lab: Focal laser stimulation of fly nociceptors activates distinct axonal and dendritic Ca 2+ signals by Basak, Sutradhar and Howard (PMID: 34175294, DOI: 10.1016/j.bpj.2021.06.001). This paper shows that axonal activation of the c4da neurons can happen even in the absence of any dendritic calcium signals. This argues that any active properties of the dendrites only occur with very strong activation. In the methodology, you should provide details about the creation of the piezo-GFP line (along with better imaging of its distribution in the dendrites). You should also add missing information about the regions of interest in which calcium responses were measured.

*Reviewer #1 (Recommendations for the authors):*

Liu et al. present findings that significantly extend the understanding of molecular and cellular pathways of mechanical nociception in *Drosophila* larvae. They present a detailed analysis of the mechanical response properties of the nociceptors that is performed using a newly developed preparation for optical recordings from these cells. Using mechanical probes of varying tip diameters they are able to investigate the responses as they relate to force and pressure. Mutants in the cut gene, which show reduced branching of the nociceptors, are used to interrogate how the dendritic morphology structure might relate to their physiological responses. As well, the response profiles of the neurons that are mutant for mechanosensory channels ppk-26 and piezo are investigated. The ppk-26 mutant shows a more strongly impaired deficit in comparison to piezo mutants and double mutants show a nearly abolished mechanical response. The mechanosensory neurons are also found to respond to mechanical forces that are outside of the main dendritic fields which suggests that they are able to detect forces that are viscoelatically coupled through the overlying epidermis. Finally, a voltage activated calcium channel is found to be important which suggests that the dendrites of the nociceptive neurons are "active" (as opposed to passive). Overall, the study significantly advances our understanding of the response properties of the mechanical nociceptors of the *Drosophila* larva.

Questions for the authors:

The duration of the calcium responses long outlasts the force application. It has been previously proposed that dendritic breakage could be a contributing factor in the transduction mechanism in the *Drosophila* nociceptive neurons (Tracey, 2017). What is the relationship of dendritic breakage occur with the various force stimuli and probe sizes in the imaging setup? Sharp probes (smaller probes) may be more likely to break dendrites. Similarly, do the sharp probes that trigger rolling in Figure 2J break dendrites?

The results of experiments with cut are very interesting, and the authors are open and cautious in their interpretation when they state the caveats of unknown epistatic effects that may result from removal of a transcription factor. Even with the stated caveats, the unknown effect of cut removal causes the results to be very difficult to interpret. It cannot be concluded that the deficits are a consequence of reduced branching. For example, perhaps calcium-alpha1d is a target.

Equation 3 assumes that the glass rod is completely rigid, if the rod bends during the stimulus then then d becomes on overestimate. Does the glass rod bend in this setup?

Are force measurements in figure 1i made with a larval filet in the setup? Or is this the force applied to PDMS alone?

Where are the values for the elastic modulus for larval cuticle coming from in line 543? Experimental measurements of the stiffness of larval cuticle come in at 0.39 +/- 0.01 MPa which is an order of magnitude lower than the values input to the model (10Mpa). Kohane M, Daugela A, Kutomi H, Charlson L, Wyrobek A, Wyrobek J. Nanoscale in vivo evaluation of the stiffness of *Drosophila melanogaster* integument during development". J Biomed Mater Res A. 2003 Sep 1;66(3):633-42. doi: 10.1002/jbm.a.10028

Third instar larval cuticle is closer to 20 microns thick according to experimental measurements. (Christine E. Kaznowski, Howard A. Schneiderman, Peter J. Bryant, "Cuticle secretion during larval growth in *Drosophila melanogaster*", Journal of Insect Physiology, Volume 31, Issue 10, 1985, Pages 801-813, https://doi.org/10.1016/0022-1910(85)90073-3).

The reference provided for elastic modulus of muscle comes from a study on human muscles and is therefore not valid for a study performed on *Drosophila* larvae.

Is the force probe is compressing the entire larval filet such that there is an indentation into the PDMS (as depicted in figure 1A)? If so, then isn't it true that the forces on the filet (dendrites) are not coming solely from the force probe itself? If the larva is being squished between the probe and the PDMS, then the larva is also being exposed to an opposing force that is coming from the PDMS.

Figure 4D. Why "dendritic coverage threshold" instead of distance from dendrite? How do the predictions from this figure relate to real response data?

Could differences in distribution of Ca-alpha1D along the dendrite explain differences in proximal/distal sensitivities (ie. inhibition has less effect at proximal, maybe channels are less expressed/less important there?) Checking Ca-alpha1D localization would seem to be important.

Figure 6A, the top soma trace seems really low, compared to the responses shown earlier with this probe size and force.

Please provide details of the creation of the piezo::GFP line and better imaging of the distribution in the dendrites.

What is the frame rate of image acquisition? Are data from a single z-slice? Is pinhole wide open?

In Line 75 please do not refer to flies and worms as "lower animals." Recommend substituting a term such as "invertebrates. "

In line 85 cite Zhong et al. 2010.

In lines 152-153 it is stated that the c4da neurons use their entire dendritic field as the force-receptive field. However, the later experiments show that forces are detected through the viscoelastic coupling of the dendrites to the rest of the epidermis therefore this statement is not supported.

In line 163 Cite Cynthia Hughes et al. and he et al.

In line 166 please cite Tsubouchi et al. 2012 (which precedes Yan et al. 2013 in the demonstration of the gentle touch function for cIIIda neurons).

In Line 201 cite Robertson JL et al. 2013.

In line 296 cite Zhong et al. 2010, Mauthner et al. 2014, Gorczyca DA et al. 2014.

In line 385 calcium α 1- D is more precisely described as a calcium channel subunit.

Figure 1a cuticle is misspelled.

*Reviewer #2 (Recommendations for the authors):*

1. The citations are in some instances used somewhat loosely and do not appropriately reflect the literature. This should be amended in a revised version:

Lines 88-89: The selection of references cited for the downstream circuitry of c4da neurons seems random. Grueber et al. 2007 characterized axonal projections of c4da only, Burgos et al. and Yoshino et al. each described a class of c4da second order neurons that have not been characterized in regard to mechanical stimuli. If the authors wish to cite all studies then critical ones are missing including Ohyama et al. (Nature 2015), Kaneko et al. (Neuron 2017), Hu et al. (Nat Neurosci 2017) and Takagi et al. (Neuron 2017). The latter two studies actually characterized the c4da circuitry in regard to noxious mechanical stimuli, which seems most relevant here. Alternatively, Gerhard et al. (*eLife* 2017) provides the most complete description of the c4da second order circuitry.

Lines 201 and 230: Grueber et al. (Cell 2003) was the first study to show that cut determines c4da neuron morphology and should be cited here.

2. Some discussion is warranted regarding the force used here (<5 mN) compared to the 5-10x higher forces typically used in previous publications studying rolling responses in live larvae (e.g. Kim et al. Nature 2012, Zhong et al. Curr. Biol. 2012, Gorczyca et al. Cell Rep. 2014, Guo et al. Cell Rep. 2014, Hu et al. Nat Neurosci 2017, Lopez-Bellido J Neurosci. 2019). While this is likely in part linked to the small diameter probes here vs. the typically larger probes utilized in previous studies, both the neuronal as well as behavioral responses to low mN stimuli seem to be extremely strong. For the neuronal activation, this might be linked to several features unique to the semi-intact preparation used by the authors vs. an intact animal. For example, the PDMS stiffness is much higher than that of a larval body filled with hemolymph, which will resist force manipulation much less, i.e. indentation depth is likely higher for a given force/pressure. This point should be adequately discussed and put into context with the literature.

3. Relating to point 2, measuring larval rolling behavior using their different sized probes is interesting, yet the methodology was not so clearly explained in the methods or results. The authors indicate that the probes exerted a force between 10-20 mN, but do not indicate if the variability of individual probes depends on the diameter of the glass probe. As the pressure on the animal depends on probe size it might make sense to check if the measured force is dependent on the probe diameter and/or if response rates are just variable within the same range independently of the probe size.

4. Figure 2J: The legend is a bit confusing as the categories listed are <1, >1 and >2 turns. Previous studies typically identified a single roll while multiple rolling indicates severe tissue damage. Some of the tissue damage effects by different probe sizes and pressures have been studied in more detail suggesting that tissue damage is actually required for defensive behavior (Lopez-Bellido et al. J Neurosci 2019 and J Vis Exp 2020). The authors should check for tissue damage under their experimental conditions to exclude that excessive rupturing of the epidermis causes the observed strong behavioral reactions.

5. In line 195 and 717, the labeling for this panel should be "J" instead of G.

6. It should be considered that the relative contributions of ppk/ppk26 and piezo might not exclusively reflect their function in c4da neurons alone. While ppk/ppk26 expression is cell type-specific, piezo is widely expressed in multiple tissues. Using a piezoko allele thus does not allow to draw conclusions on its function in c4da neurons. This would require either a cell type-specific manipulation or a c4da neuron specific rescue experiment. It would help if the authors can model lateral tension and perpendicular pressure for the 30 micron probe. This might help to explain the different defects described in ppk26 vs. piezo loss of function. Correlating the differential reliance of ppk26 and piezo on Ca-alpha1D function might be another interesting aspect, given that distal function of piezo to some extent resembles the one of Ca-alpha1D.

7. The model in Figure 7 could be improved as the color code and labels are not intuitive and the assignment of piezo vs ppk function is not clear to me. The model might also have to be revised according to the criticism regarding the assigned functions of ppk26 vs piezo.

*Reviewer #3 (Recommendations for the authors):*

– Figure 2, legend: (G) should read (J).

– Line 160: "To further justify the validity" – change to "To further test the validity".

– Line 66: not sure whether e.g. proprioceptors controlling locomotion aren't equally important for the survival of an organism.

– Line 390 ff: use past tense when reporting on earlier findings.

---

## [Author Response]

The reviewers have discussed at length their reviews with one another, and the Reviewing Editor has drafted this to help you prepare a revised submission. The reviewers were generally very impressed by the quality of your study and the novelty of your results. Nonetheless, a series of significant technical concerns were raised. We are confident that you will be able to address these concerns, and ask that you do so in a revised manuscript. Although the proposed revisions will entail additional work and new data, we encourage you to build on existing results. Specially:1. Please check if dendrite/epidermal breakage occurs with the used probes. Ruling out such breakages is important to evaluate the data and its interpretation both at the level of the calcium signals and behavior. You should use ppk-tdtomato or ppk-mCD8GFP for this to make sure dendrites are bright enough and visible. Also, magnification higher than 20X is probably needed.

We thank the reviewers for this suggestion. In our experiments, we always recorded the dendritic morphology before and after the force stimulation to control dendritic breakage. To make this point clearer, we made two changes in the revised manuscript. First, in the revised Figure 1—figure supplement 1, we showed that in the condition of using a saturation force, no significant damage of dendritic morphology was observed. Second, we made it clearer now that it was our intention to stay with the small deformation constrain to focus on mechanosensory responses of c4da (see page 5, second paragraph, line 124-130).

2. Please use the existing set of recordings to evaluate the possibility of dendritic calcium signals. The idea is to analyze the spatial distribution of dendritic calcium responses to the pokes. If no or very weak dendritic calcium signals are found, this observation should be stated in the manuscript. If significant dendritic calcium signals are observed, you should determine the dependence of these signals on Ca-alpha1D.

In the original manuscript, we included the data on the dendritic calcium signal and showed that the dendritic signal was reduced when the activity of VGCCs were inhibited or in the Ca-α1D knockdown mutant (see Figure 6 A-B in the original manuscript). Inspired by the suggestion from the reviewers, we had a closer look at our data and performed additional experiments. In the revised Figure 6 A-B, we showed that the mechanical stimuli could evoke calcium responses not only in the soma, but also in the homolateral (i.e. between the soma and the force probe) and contralateral (i.e. opposite side of the force probe) dendrites, suggesting that the dendritic signals are propagating within the dendritic arbors. Moreover, in the revised Figure 6 A-B and Figure 6—figure supplement 1, we showed that these dendritic signals were reduced in the mutant strains of *Ca-α1D* or if the fillet preparation was treated with nimodipine, demonstrating a clear dependence on the activity of VGCCs. However, because our imaging speed is not fast enough to capture the dendritic flow of calcium signals, the dynamics of signal propagation remains undefined. This would be an interesting issue to study in the future. Along with the revised Figure 6, we also revised the text and legends accordingly.

3. Please validate the RNAi knockdown and mutant results, particularly for Ca-alpha1D and piezo. For the RNAi lines used, qPCR data on the knockdown-efficiency should be added. If the qPCR do not work, you should consider using piezo-GFP as a readout for assessing the knockdown efficiency.

First, we did not use RNAi mutant for Piezo. The *Piezo*^KO^ line is a genomic mutant strain^1^.

Second, for *Ca-α1D*, because there are only a small number of c4da in each animal and Ca-α1D has a quite broad expression in various types of neurons (see our revised Figure 6—figure supplement 2), we expected that the reduction in the expression level of *Ca-α1D* in c4da would be very difficult to detect. Therefore, we knocked down the expression of *Ca-α1D* in the whole animal using the same *uas-Ca-α1D^i^* strain and the *tub-gal4* strain. Using RT-PCR, we showed that the expression level of *Ca-α1D* was significantly reduced (revised Figure 6—figure supplement 2). In fact, the same RNAi strain was also used in other functional studies^2,3^.

For Ca-alpha1D, you should check the relevant mutants that could be used to provide better genetic evidence in support of a role for Ca-alpha1D.

We thank the reviewers for this suggestion. We have now performed additional functional experiments on *Ca-α1D*^X10/AR66^, a genomic mutant strain used in previous studies on Ca-α1D^2,4^. As said by the reviewer, the additional data strengthens the genetic evidence to support the role of Ca-α1D.

Data on protein localization is presented for the mechanosensitive channels, but not for Ca-alpha1D. Knowing the distribution of the Ca-alpha1D subunit is critical to the interpretation of the rest of your data. To do this, we recommend you use an antibody against Ca-alpha1D that was generated in I-Uen Hsu et al., PNAS 2020, PMID:33168737. Confirming expression, localization and RNAi knockdown should be feasible if the antibody proves to be specific enough. If the antibody's specificity is insufficient, you should check whether Ca-alpha1D is expressed in the cell using e.g. Mi{ET1}Ca-α1D[MB06807] that is available at Bloomington. For mutant alleles of Ca-alpha1D, combinations of available reagents were used in a previous study cited by the author (Terada et al. ELife 2016): Ca-α1DX7/AR66 and Ca-α1DX10/AR66. While we realize these extra experiments represent substantial work, their results will be important to clarify the role of Ca-alpha1D and piezo to support several or your statements. In case the experiments listed above are unsuccessful, we ask you to tone down the interpretation of your results, which would be unfortunate.

We thank the reviewers for these suggestions and agree that it is important to determine the cell biological properties of Ca-α1D. We performed all suggested experiments and additional ones to examine the expression and localization of Ca-α1D. However, most of the results are unfortunately negative, and we now summarize these results.

First, we performed the expression analysis using the existing RNA sequencing data. Our analysis showed that *Ca-α1D* had a broad expression in various types of neurons, including those expressing *ppk1*/*ppk26* (e.g. c4da) (Figure 6—figure supplement 2). From the sequencing data, we noted that the expression level of *Ca-α1D* was generally low (Figure 6—figure supplement 2). To further confirm the expression of Ca-α1D in c4da, we used the *Mi{ET1}Ca-α1D*^MB06807^ line as suggested by the reviewers. However, no clear signal was observed.

Second, we generated the transgenic strain that could express mcherry-Ca-α1D under the endogenous promoter of *Ca-α1D*. However, no clear signal was observed. We also used the antibody as suggested by the reviewer ^5^ and no clear signal was observed in any part of c4da. We thought that it might be due to the low expression level of Ca-α1D, so we generated a *uas-mcherry-Ca-α1D* strain. Using the c4da specific ppk-gal4 strain to drive the expression of mCherry-Ca-α1D, we still could not see any signals. Because Ca-α1D is a subunit of the voltage-gated channel, we speculate that mCherry-Ca-α1D is probably not stable or is not able to assemble with the other subunits to form a functional VGCC. Future work is required to explore a good construct or other strategies to clarify the subcellular localization of *Ca-α1D*.

In all, although the cell biological evidence is lacking, we think that there is still good evidence from sequencing analysis and functional experiments to support the expression and the role of Ca-α1D in c4da. We have now summarized all our analysis of Ca-α1D in the revised manuscript (Figure 6—figure supplement 2), including all negative results, which would be useful not only as a set of supporting data but also for the information of other researchers. Finally, we also have added discussions on this issue and modified the text to ensure that the interpretation on the role of Ca-α1D is precise (see page 10, fourth paragraph, line 340-352, page 11, second paragraph, line 369-392 and the revised Discussion).

4. Please check/redo the calculations related the elastic modulus (PDMS vs. larval cuticle) and clarify the differences in the mechanical properties of the open filet preparation vs. the intact animal. In particular, transferring the elastic modulus of muscles characterized in humans to flies does not seem to be legitimate. The modeling should be repeated with a realistic value for the elastic modulus of the larva, along with an improved discussion of how the force distributions would compare to those in an intact larva. In the discussion, you should address the apparent discrepancy between the force used in your work (<5 mN) compared to the 5-10x higher forces typically used in previous publications studying rolling responses in live larvae.

First, we have now added a paragraph in the Discussion to clarify the mechanical differences between the semi-intact fillet preparation and the intact animal, in which we also discuss the possible reason underlying the discrepancy between the forces used in different studies (see Discussion, second paragraph, page 11). Second, we have checked the values for elastic modulus of insect muscle cells^6^, and it is similar to that of human muscle cells. We have now changed the reference for this point in the Methods. Moreover, because the muscle layer is expected to make little contribution to the force distribution, it is actually omitted in our simulations. Third, after looking into the literatures on the cuticle stiffness (including the one that the reviewer suggested), we think that there was still uncertainty on the modulus of insect cuticle (see below our discussion in the responses to the suggestion of reviewer 1). Therefore, we explored the parameter space by performing additional simulations using a lower value for the cuticle modulus (1 MPa). We showed that whether this modulus being 1 or 10 MPa had no effects on our conclusions. The new simulation results (*E*_cuticle_=1 MPa) were now included in Figure 4—figure supplement 1 for comparison.

5. Please provide a minimal set of controls for behavioral data. This should include the driver control for the RNAi, the driver alone, and the driver plus RNAi. You have conducted these controls for the experiments related to cut but you are missing driver alone for Ca-alpha1D.

All checked and the missing controls were added.

6. Please address the points raised by the reviewers regarding the interpretation of the results, the statistics and grammatical errors. Among the missing references, it would be important to incorporate into the discussion the following paper from the Howard lab: Focal laser stimulation of fly nociceptors activates distinct axonal and dendritic Ca 2+ signals by Basak, Sutradhar and Howard (PMID: 34175294, DOI: 10.1016/j.bpj.2021.06.001). This paper shows that axonal activation of the c4da neurons can happen even in the absence of any dendritic calcium signals. This argues that any active properties of the dendrites only occur with very strong activation.

We thank the reviewers for all the suggestions. We have tried our best to improve the manuscript in these aspects. In particular, we have now added a paragraph to incorporate the discussion on the paper from the Howard lab (see Discussion, Line 457-481).

In the methodology, you should provide details about the creation of the piezo-GFP line (along with better imaging of its distribution in the dendrites). You should also add missing information about the regions of interest in which calcium responses were measured.

The Piezo-GFP line has been published^7^. We have now added information about how and where we measured the calcium responses in the figure legends.

Reviewer #1 (Recommendations for the authors):Liu et al. present findings that significantly extend the understanding of molecular and cellular pathways of mechanical nociception in *Drosophila* larvae. They present a detailed analysis of the mechanical response properties of the nociceptors that is performed using a newly developed preparation for optical recordings from these cells. Using mechanical probes of varying tip diameters they are able to investigate the responses as they relate to force and pressure. Mutants in the cut gene, which show reduced branching of the nociceptors, are used to interrogate how the dendritic morphology structure might relate to their physiological responses. As well, the response profiles of the neurons that are mutant for mechanosensory channels ppk-26 and piezo are investigated. The ppk-26 mutant shows a more strongly impaired deficit in comparison to piezo mutants and double mutants show a nearly abolished mechanical response. The mechanosensory neurons are also found to respond to mechanical forces that are outside of the main dendritic fields which suggests that they are able to detect forces that are viscoelatically coupled through the overlying epidermis. Finally, a voltage activated calcium channel is found to be important which suggests that the dendrites of the nociceptive neurons are "active" (as opposed to passive). Overall, the study significantly advances our understanding of the response properties of the mechanical nociceptors of the *Drosophila* larva.Questions for the authors:The duration of the calcium responses long outlasts the force application. It has been previously proposed that dendritic breakage could be a contributing factor in the transduction mechanism in the *Drosophila* nociceptive neurons (Tracey, 2017). What is the relationship of dendritic breakage occur with the various force stimuli and probe sizes in the imaging setup? Sharp probes (smaller probes) may be more likely to break dendrites. Similarly, do the sharp probes that trigger rolling in Figure 2J break dendrites?

Please see our response to the first point in the above Editor’s summary. Briefly, within the range of forces and probe sizes used in this study, we did not observe significant dendrite breakage in both calcium imaging experiments and in the rolling behavior assays. We have now added data to make this point clearer (see Figure 1—figure supplement 1 in the revised manuscript). In addition, we showed that the responses of c4da to the force probes depends on Piezo and ppk1/ppk26, suggesting that mechanosensation, rather than chemosensation, is the major pathway underlying the calcium responses observed in this work.

We also noted that the duration of calcium responses was longer than that of force application. Our preliminary data suggest that intracellular calcium release (e.g. from ER) contributes to the calcium signal after the force application. The key molecule in this process is the ryanodine receptors (RyRs) in the membrane of ER. Drug treatment or knocking down the expression of the RyRs in c4da significantly could reduce the amplitude and duration of the force-evoked calcium responses (data not shown). This issue is currently under further investigation and we hope to clarify it in the near future.

The results of experiments with cut are very interesting, and the authors are open and cautious in their interpretation when they state the caveats of unknown epistatic effects that may result from removal of a transcription factor. Even with the stated caveats, the unknown effect of cut removal causes the results to be very difficult to interpret. It cannot be concluded that the deficits are a consequence of reduced branching. For example, perhaps calcium-alpha1d is a target.

We agree with the reviewer that the potential contribution of the unknown effect as a result of cut knockdown cannot be absolutely excluded. We now made this point clear in the revised Results and Discussion. On the other hand, the observations on the *cut* mutant are consistent with idea of the dendritic complexity being a contributing parameter underlying the signal-encoding mechanism of c4da. This argument is also supported by the observations in c3da. We have made open discussion on this issue (see Discussion, page 14-15).

Equation 3 assumes that the glass rod is completely rigid, if the rod bends during the stimulus then then d becomes on overestimate. Does the glass rod bend in this setup?

No. The glass probe does not bend as it is very short and has a strong base (see Author response image 1).

**Author response image 1. sa2fig1:** The schematics for how the probe was mounted. As shown in the picture, the force probe was firmly mounted in the double-layered holder. These glass probes were short and solid, so they would not be bent during the experiments.

Are force measurements in figure 1i made with a larval filet in the setup? Or is this the force applied to PDMS alone?

The forces were measured in the presence of a larval fillet.

Where are the values for the elastic modulus for larval cuticle coming from in line 543? Experimental measurements of the stiffness of larval cuticle come in at 0.39 +/- 0.01 MPa which is an order of magnitude lower than the values input to the model (10Mpa). Kohane M, Daugela A, Kutomi H, Charlson L, Wyrobek A, Wyrobek J. Nanoscale in vivo evaluation of the stiffness of *Drosophila melanogaster* integument during development". J Biomed Mater Res A. 2003 Sep 1;66(3):633-42. doi: 10.1002/jbm.a.10028

We think that the cuticle stiffness reported in the suggested literatures still has uncertainty for technical reasons. First, the two ends of the larvae were not fixed during the force application, which would lead to underestimation of cuticle stiffness. Second, the mechanical contribution of internal liquid pressure and the rigidity of the muscle layer are difficult to be separated from that of cuticle, which may lead to overestimation of cuticle stiffness. Based on these considerations, we decided to explore the parameter space of *E*_cuticle_ by performing additional simulations using a lower value for the cuticle modulus (1 MPa). We showed that whether this modulus being 1 or 10 MPa had no effect on our conclusions because the cuticle is thin in comparison to the PDMS pad. The new simulation results (*E*_cuticle_=1 MPa) were now included in Figure 4—figure supplement 1 for comparison. Furthermore, we have added discussion on the mechanical differences between our preparation and the intact animals (see Discussion, second paragraph).

Third instar larval cuticle is closer to 20 microns thick according to experimental measurements. (Christine E. Kaznowski, Howard A. Schneiderman, Peter J. Bryant, "Cuticle secretion during larval growth in *Drosophila melanogaster*", Journal of Insect Physiology, Volume 31, Issue 10, 1985, Pages 801-813, https://doi.org/10.1016/0022-1910(85)90073-3).

10 μm is within the range of larval cuticle thickness based on our own EM observations (see Author response image 2) and others’ work, such as ^8^.

**Author response image 2. sa2fig2:** Representative TEM micrographs on *Drosophila* larval fillet preparation. The thickness of the cuticle was indicated. The samples were prepared using high pressure freezing and freeze substitution as previously described^9^.

The reference provided for elastic modulus of muscle comes from a study on human muscles and is therefore not valid for a study performed on *Drosophila* larvae.

We thank the reviewer for pointing out this mistake. We checked the values for the elastic modulus of insect muscle cells^6^, and it is similar to that of human cells. We have now changed the reference accordingly. It does not affect our simulation results because the muscle layer is expected to make little contribution to the force distribution, it was actually omitted in our simulations.

Is the force probe is compressing the entire larval filet such that there is an indentation into the PDMS (as depicted in figure 1A)? If so, then isn't it true that the forces on the filet (dendrites) are not coming solely from the force probe itself? If the larva is being squished between the probe and the PDMS, then the larva is also being exposed to an opposing force that is coming from the PDMS.

Yes, there is also a force from the PDMS. In physiological conditions, when the mechanical poking is applied to the larva, c4da also withstand forces from the other tissues, such as the muscle layers. The difference is that the rigidities of PDMS and the muscle layer are different and thereby would have different contributions to the overall tissue mechanics. We have now added discussions on the mechanical differences between our fillet preparation and the intact animals (see Discussion, second paragraph).

Figure 4D. Why "dendritic coverage threshold" instead of distance from dendrite? How do the predictions from this figure relate to real response data?

In our simulations, we tested different values for *C*_d_ (i.e. dendritic coverage threshold) and other parameters (data not shown) to explore the parameter space of our model. Figure 4D showed how the simulated probability of activation changes with the value of *C*_d_. We think that this helps to understand how *T*_L_ promotes the probability of neuronal activation and the robustness of this parameter in the model.

Regarding how cell activation changes with the distance from a dendrite, we have shown that the mechanosensitivity to *T*_L_ would expand the radius of the force-receptive field of c4da for a few tens of micrometers (Figure 4B). When the probability of cell activation is calculated and plotted against the distance of force from a dendrite, similar conclusion can be made (Author response image 3). Both are consistent with the experimental observations (Figure 4A).

**Author response image 3. sa2fig3:** The mechanosensitivity to lateral tension increases the probability of cell activation. (A) The representative color map for the distance to the nearest dendrite, in which the distance was labeled by different colors. Scale bar, 100 μm. Random positions were chosen (the red crosses) as the force application points in the simulations (*n*=5 cells, 30 positions for each cell). (B-C) The plots of the cell activation probability versus the distance from a dendrite. Two types of probes were simulated, i.e. 30 μm (B) and 60 μm (C). Three conditions were considered: (1) only sensitive to *P*_P_; (2) only sensitive to *T*_L_; (3) sensitive to both *P*_P_ and *T*_L_. The dendritic coverage threshold set was 20 μm. In panel B and C, data were presented as mean±std.

Could differences in distribution of Ca-alpha1D along the dendrite explain differences in proximal/distal sensitivities (ie. inhibition has less effect at proximal, maybe channels are less expressed/less important there?) Checking Ca-alpha1D localization would seem to be important.

Please see our response to the third point in the above Editor’s summary.

Figure 6A, the top soma trace seems really low, compared to the responses shown earlier with this probe size and force.

We checked the data and found that this trace was still within the range of responses. We now changed the images and curves to be more representative (see revised Figure 6A).

Please provide details of the creation of the piezo::GFP line and better imaging of the distribution in the dendrites.

The information about this strain has been published ^7^. We have now included this reference. This Piezo-GFP line is a knock-in strain, so the signal is quite weak. The image shown now in the revised manuscript is a representative one.

What is the frame rate of image acquisition? Are data from a single z-slice? Is pinhole wide open?

The frame rate is about 7.7 fps (130 ms per frame) and the images were taken using the confocal mode. The soma and dendrite usually get back to original vertical position with a brief latency (usually 1-2 s), which is shorter than the rising time of the calcium responses (~10 s).

In Line 75 please do not refer to flies and worms as "lower animals." Recommend substituting a term such as "invertebrates. "

Corrected.

In lines 152-153 it is stated that the c4da neurons use their entire dendritic field as the force-receptive field. However, the later experiments show that forces are detected through the viscoelastic coupling of the dendrites to the rest of the epidermis therefore this statement is not supported.

We have checked this issue throughout the revised manuscript and modified the text accordingly to be more precise.

In line 85 cite Zhong et al. 2010.In line 163 Cite Cynthia Hughes et al. and he et al.In line 166 please cite Tsubouchi et al. 2012 (which precedes Yan et al. 2013 in the demonstration of the gentle touch function for cIIIda neurons).In Line 201 cite Robertson JL et al. 2013.In line 296 cite Zhong et al. 2010, Mauthner et al. 2014, Gorczyca DA et al. 2014.

All cited. We thank the reviewer for these suggestions.

In line 385 calcium α 1- D is more precisely described as a calcium channel subunit.

Corrected.

Figure 1a cuticle is misspelled.

Corrected.

Reviewer #2 (Recommendations for the authors):1. The citations are in some instances used somewhat loosely and do not appropriately reflect the literature. This should be amended in a revised version:Lines 88-89: The selection of references cited for the downstream circuitry of c4da neurons seems random. Grueber et al. 2007 characterized axonal projections of c4da only, Burgos et al. and Yoshino et al. each described a class of c4da second order neurons that have not been characterized in regard to mechanical stimuli. If the authors wish to cite all studies then critical ones are missing including Ohyama et al. (Nature 2015), Kaneko et al. (Neuron 2017), Hu et al. (Nat Neurosci 2017) and Takagi et al. (Neuron 2017). The latter two studies actually characterized the c4da circuitry in regard to noxious mechanical stimuli, which seems most relevant here. Alternatively, Gerhard et al. (eLife 2017) provides the most complete description of the c4da second order circuitry.Lines 201 and 230: Grueber et al. (Cell 2003) was the first study to show that cut determines c4da neuron morphology and should be cited here.

Corrected. We thank the reviewer for these suggestions.

2. Some discussion is warranted regarding the force used here (<5 mN) compared to the 5-10x higher forces typically used in previous publications studying rolling responses in live larvae (e.g. Kim et al. Nature 2012, Zhong et al. Curr. Biol. 2012, Gorczyca et al. Cell Rep. 2014, Guo et al. Cell Rep. 2014, Hu et al. Nat Neurosci 2017, Lopez-Bellido J Neurosci. 2019). While this is likely in part linked to the small diameter probes here vs. the typically larger probes utilized in previous studies, both the neuronal as well as behavioral responses to low mN stimuli seem to be extremely strong. For the neuronal activation, this might be linked to several features unique to the semi-intact preparation used by the authors vs. an intact animal. For example, the PDMS stiffness is much higher than that of a larval body filled with hemolymph, which will resist force manipulation much less, i.e. indentation depth is likely higher for a given force/pressure. This point should be adequately discussed and put into context with the literature.

Please see our response to the fourth point in the Editor’s summary. We have now added a paragraph in the Discussion (see Discussion, second paragraph) to clarify the mechanical differences between the semi-intact fillet preparation and the intact animal, including the discussion on the smaller forces used in the present study.

3. Relating to point 2, measuring larval rolling behavior using their different sized probes is interesting, yet the methodology was not so clearly explained in the methods or results. The authors indicate that the probes exerted a force between 10-20 mN, but do not indicate if the variability of individual probes depends on the diameter of the glass probe. As the pressure on the animal depends on probe size it might make sense to check if the measured force is dependent on the probe diameter and/or if response rates are just variable within the same range independently of the probe size.

We have now modified the corresponding part in Methods. We now included the data to show that the stimulating force was independent on the probe size (Figure 2—figure supplement 3 and page 6, line 189-191). Because the glass probes were all very short, they would not be deformed during the poking experiments. Therefore, all forces are reflected in the form of the bending of the Von Frey fiber, which are the same for all the probes.

4. Figure 2J: The legend is a bit confusing as the categories listed are <1, >1 and >2 turns. Previous studies typically identified a single roll while multiple rolling indicates severe tissue damage. Some of the tissue damage effects by different probe sizes and pressures have been studied in more detail suggesting that tissue damage is actually required for defensive behavior (Lopez-Bellido et al. J Neurosci 2019 and J Vis Exp 2020). The authors should check for tissue damage under their experimental conditions to exclude that excessive rupturing of the epidermis causes the observed strong behavioral reactions.

We have now used the same way as the previous studies to quantify the behavior responses. We have accordingly modified the figures and the texts in the Methods. Furthermore, we have checked the dendritic damage for both imaging and behavior experiments. Within the range of forces and probe sizes used in this study, we did not observe significant dendrite breakage (see Figure 1—figure supplement 1 in the revised manuscript).

5. In line 195 and 717, the labeling for this panel should be "J" instead of G.

Corrected.

6. It should be considered that the relative contributions of ppk/ppk26 and piezo might not exclusively reflect their function in c4da neurons alone. While ppk/ppk26 expression is cell type-specific, piezo is widely expressed in multiple tissues. Using a piezoko allele thus does not allow to draw conclusions on its function in c4da neurons. This would require either a cell type-specific manipulation or a c4da neuron specific rescue experiment.

We thank the reviewer for this suggestion. We have now performed c4da-specific rescue experiments for *piezo* to strengthen our conclusions (see revised Figure 5G and page 9, line 315-328).

It would help if the authors can model lateral tension and perpendicular pressure for the 30 micron probe. This might help to explain the different defects described in ppk26 vs. piezo loss of function.

We thank the reviewer for this suggestion. We have performed the additional simulations and the new data were now in the revised Figure 4.

Correlating the differential reliance of ppk26 and piezo on Ca-alpha1D function might be another interesting aspect, given that distal function of piezo to some extent resembles the one of Ca-alpha1D.

We thank the reviewer for the suggestion and feel that this is a very interesting issue to follow up. One key piece of information to understand this issue is the subcellular localization of Ca-α1D and the mutual dependence between Piezo and Ca-α1D. However, as we described in the responses to Editor’s summary, the subcellular localization of Ca-α1D in c4da was still not clear. Therefore, this issue would need to be further explored in future studies.

7. The model in Figure 7 could be improved as the color code and labels are not intuitive and the assignment of piezo vs ppk function is not clear to me. The model might also have to be revised according to the criticism regarding the assigned functions of ppk26 vs piezo.

In this model figure, what we would like to point out is that both Piezo and ppk1/ppk26 contribute to the mechanosensitivity to lateral tension. To make this point clear, we have modified this figure. In particular, we simplified the colors to avoid misunderstanding. We also edited the legends and the corresponding texts in Discussion accordingly. With respect to the roles of Piezo and ppk1/ppk26, we discussed their differential contributions in the Discussion.

Reviewer #3 (Recommendations for the authors):– Figure 2, legend: (G) should read (J).

Corrected.

– Line 160: "To further justify the validity" – change to "To further test the validity".

Corrected.

– Line 66: not sure whether e.g. proprioceptors controlling locomotion aren't equally important for the survival of an organism.

Edited to be more precise.

– Line 390 ff: use past tense when reporting on earlier findings.

Checked.

References

1 Kim, S. E., Coste, B., Chadha, A., Cook, B. & Patapoutian, A. The role of *Drosophila* Piezo in mechanical nociception. *Nature* 483, 209-212, doi:10.1038/nature10801 (2012).

2 Terada, S. *et al.* Neuronal processing of noxious thermal stimuli mediated by dendritic Ca(2+) influx in *Drosophila* somatosensory neurons. *Elife* 5, doi:10.7554/eLife.12959 (2016).

3 Basak, R., Sutradhar, S. f. C. U. X. L. D.-s.-S.-m. p. & Howard, J. Focal laser stimulation of fly nociceptors activates distinct axonal and dendritic Ca(2+) signals. *Biophys J* 120, 3222-3233, doi:10.1016/j.bpj.2021.06.001 (2021).

4 Eberl, D. F. *et al.* Genetic and developmental characterization of Dmca1D, a calcium channel alpha1 subunit gene in *Drosophila melanogaster*. *Genetics* 148, 1159-1169, doi:10.1093/genetics/148.3.1159 (1998).

5 Hsu, I. U. *et al.* Stac protein regulates release of neuropeptides. *Proc Natl Acad Sci U S A* 117, 29914-29924, doi:10.1073/pnas.2009224117 (2020).

6 Zhang, Y., Pang, X., Yang, Y. & Yan, S. Effect of calcium ion on the morphology structure and compression elasticity of muscle fibers from honeybee abdomen. *J Biomech* 127, 110652, doi:10.1016/j.jbiomech.2021.110652 (2021).

7 Song, Y. *et al.* The Mechanosensitive Ion Channel Piezo Inhibits Axon Regeneration. *Neuron* 102, 373-389 e376, doi:10.1016/j.neuron.2019.01.050 (2019).

8 Kohane, M. *et al.* Nanoscale in vivo evaluation of the stiffness of *Drosophila melanogaster* integument during development. *J Biomed Mater Res A* 66, 633-642, doi:10.1002/jbm.a.10028 (2003).

9 Sun, L. *et al.* Ultrastructural organization of NompC in the mechanoreceptive organelle of *Drosophila* campaniform mechanoreceptors. *Proc Natl Acad Sci U S A* 116, 7343-7352, doi:10.1073/pnas.1819371116 (2019).